# Cell-type diversity and regionalized gene expression in the planarian intestine

**David J Forsthoefel[1,2]\*, Nicholas I Cejda[1], Umair W Khan[2†], Phillip A Newmark[2‡]\***

[1]Genes and Human Disease Research Program, Oklahoma Medical Research Foundation, Oklahoma City, United States; [2]Howard Hughes Medical Institute, Department of Cell and Developmental Biology, University of Illinois at Urbana-Champaign, Urbana, United States

**\*For correspondence:**
david-forsthoefel@omrf.org (DJF);
pnewmark@morgridge.org (PAN)

**Present address:** †Graduate Program in Cell and Molecular Biology, University of Wisconsin–Madison, Madison, United States; ‡Howard Hughes Medical Institute, Morgridge Institute for Research, Department of Integrative Biology, University of Wisconsin-Madison, Madison, United States

**Abstract** Proper function and repair of the digestive system are vital to most animals. Deciphering the mechanisms involved in these processes requires an atlas of gene expression and cell types. Here, we applied laser-capture microdissection (LCM) and RNA-seq to characterize the intestinal transcriptome of *Schmidtea mediterranea*, a planarian flatworm that can regenerate all organs, including the gut. We identified hundreds of genes with intestinal expression undetected by previous approaches. Systematic analyses revealed extensive conservation of digestive physiology and cell types with other animals, including humans. Furthermore, spatial LCM enabled us to uncover previously unappreciated regionalization of gene expression in the planarian intestine along the medio-lateral axis, especially among intestinal goblet cells. Finally, we identified two intestine-enriched transcription factors that specifically regulate regeneration (hedgehog signaling effector *gli-1*) or maintenance (*RREB2*) of goblet cells. Altogether, this work provides resources for further investigation of mechanisms involved in gastrointestinal function, repair and regeneration.

## Introduction

Physical trauma, disease, and aging can damage the digestive tract, causing numerous human gastrointestinal (GI) pathologies (*Li and Jasper, 2016*; *Andersson-Rolf et al., 2017*; *Peery et al., 2019*). Mice and *Drosophila* can repair damage to the digestive epithelium, and recent studies have elucidated cellular and molecular mechanisms underpinning these abilities (*Gehart and Clevers, 2019*; *Jiang et al., 2016*; *Zwick et al., 2019*). Some animals are endowed with much greater regenerative capacity, and can repair or even completely replace severely damaged or missing GI tissue (*Goodchild, 1956*; *O'Steen, 1958*; *Takeo et al., 2008*; *Kaneko et al., 2010*; *Zattara and Bely, 2011*; *Mashanov et al., 2014*; *Okano et al., 2015*), but the underlying mechanisms are far less understood. Regeneration requires precise spatial and temporal control over the differentiation of distinct cell types, as well as remodeling of uninjured tissue. Furthermore, individual cell types can respond uniquely to injury and play specialized roles that promote regeneration (*Gehart and Clevers, 2019*; *Jiang et al., 2016*; *Kumar et al., 2007*; *Witchley et al., 2013*; *Gemberling et al., 2015*; *Mokalled et al., 2016*; *Tanaka, 2016*). Therefore, characterization of an organ's composition and gene expression at a cellular level in the uninjured state is an essential step in unraveling the mechanisms required for faithful re-establishment of organ morphology and physiology.

Driven by the recent application of genomic and molecular methods, the planarian flatworm *Schmidtea mediterranea* has become a powerful model in which to address the molecular and cellular underpinnings of organ regeneration (*Newmark and Sánchez Alvarado, 2002*; *Robb et al., 2015*; *Brandl et al., 2016*; *Grohme et al., 2018*; *Reddien, 2018*). In response to nearly any type of surgical amputation injury, pluripotent stem cells called neoblasts proliferate and differentiate, regenerating brain, intestine, and other tissues lost to injury (*Reddien and Sánchez Alvarado, 2004*; *Rink, 2018*). In addition, pre-existing tissue undergoes extensive remodeling and re-scaling through

**eLife digest** The human body has a limited ability to regenerate and repair itself after major injuries. By contrast, flatworms – most notably planarians such as *Schmidtea mediterranea* – have exceptional regenerative abilities and can regrow large parts of their bodies. Regrowing body parts is a complex process involving the coordinated creation of many different types of cells, and thus an important first step in understanding tissue regeneration is to develop a detailed catalog of cell types in that tissue.

Laser capture microdissection, or LCM for short, is a technology used to isolate and study subregions or even individual cells from within a tissue. This approach can help to identify different cell types and to examine what makes them unique. LCM can be used to create a detailed catalog of cells, their differences and the roles they perform.

Forsthoefel et al. have now used LCM to study cells from the planarian digestive system. This approach found 1,800 genes that have high activity in cells from the gut and showed many similarities between planaria and humans. LCM made it possible to study these cells in a new level of detail, revealing several hundred new genes as well as new cell types. The study showed that regeneration and survival of cells known as goblet cells particularly depended on two genes, *gli-1* and *RREB2*.

Irreversible gut damage in humans can result from surgeries and conditions such as acid reflux. Other animals are able to repair and regenerate the gut more successfully. Techniques like LCM can help researchers to understand the differences between humans and other species. In time, these insights may lead to technologies and therapies that can improve our own abilities to heal following injuries.

both collective migration of post-mitotic cells in undamaged tissues, as well as proportional loss of cells through apoptosis (*Pellettieri, 2019*). These processes are coordinated to achieve re-establishment of proportion, symmetry, and function of planarian organ systems within a few weeks after injury (*Roberts-Galbraith and Newmark, 2015*).

The planarian intestine is a prominent organ whose highly branched morphology, simple cellular composition, and likely cell non-autonomous role in neoblast regulation make it a compelling model for addressing fundamental mechanisms of regeneration. In uninjured planarians, a single anterior and two posterior primary intestinal branches project into the head and tail, respectively, with secondary, tertiary, and quaternary branches extending toward lateral body margins (*Hyman, 1951*; *Forsthoefel et al., 2011*). Growth and regeneration of intestinal branches require considerable remodeling of pre-existing tissue (*Forsthoefel et al., 2011*). Remodeling is governed by axial polarity cues (*Pellettieri, 2019*; *Forsthoefel and Newmark, 2009*), extracellular-signal-regulated kinase (ERK) and epidermal growth factor receptor (EGFR) signaling pathways (*Umesono et al., 2013*; *Hosoda et al., 2016*; *Barberán et al., 2016*), cytoskeletal regulators (*Forsthoefel et al., 2012*), and interactions with muscle (*Adler and Sánchez Alvarado, 2017*; *Seebeck et al., 2017*; *Bonar and Petersen, 2017*; *Scimone et al., 2018*). However, the mechanisms by which post-mitotic intestinal cells sense and respond to extrinsic signals are only superficially understood.

New intestinal cells (the progeny of neoblasts) differentiate at the severed ends of injured gut branches, as well as in regions of significant remodeling, providing an intriguing example of how differentiation and remodeling must be coordinated to achieve integration of old and new tissue (*Forsthoefel et al., 2011*). Only three cell types comprise the intestinal epithelium: secretory goblet cells, absorptive phagocytes (*Willier et al., 1925*; *Ishii, 1965*; *Bowen et al., 1974*), and a recently identified population of basally located 'outer' intestinal cells (*Fincher et al., 2018*). Transcription factors expressed by intestinal progenitors ('gamma' neoblasts) and their progeny have been identified (*Forsthoefel et al., 2012*; *Fincher et al., 2018*; *Wagner et al., 2011*; *van Wolfswinkel et al., 2014*; *Labbé et al., 2012*; *Zeng et al., 2018*). However, only the EGF receptor *egfr-1* (*Barberán et al., 2016*) has been shown definitively to be required for integration of intestinal cells into gut branches, and therefore the functional requirements for differentiation of new intestinal cells are largely undefined.

The intestine may also play a niche-like role in modulating neoblast dynamics. Knockdown of several intestine-enriched transcription factors (*nkx2.2, gata4/5/6-1*) causes reduced blastema formation and/or decreased neoblast proliferation (*Forsthoefel et al., 2012*; *Flores et al., 2016*). Similarly, knockdown of the intestine-enriched HECT E3 ubiquitin ligase *wwp1* causes disruption of intestinal integrity, reduced blastema formation, and neoblast loss (*Henderson et al., 2015*). Conversely, knockdown of *egfr-1* causes hyperproliferation and expansion of several neoblast subclasses (*Barberán et al., 2016*). Because these genes are also expressed by neoblasts and their progeny, careful analysis will be required to distinguish their functions in the stem cell compartment from cell non-autonomous roles in the intestine. Nonetheless, because so few extrinsic signals (*Miller and Newmark, 2012*; *Gaviño et al., 2013*; *Dingwall and King, 2016*; *Lei et al., 2016*) controlling neoblast proliferation have been identified, further investigation of the intestine as a potential source of such cues is warranted.

Addressing these aspects of intestine regeneration and function necessitates development of approaches for purification of intestinal tissue. Previously, we developed a method for purifying intestinal phagocytes from single-cell suspensions derived from planarians fed magnetic beads, enabling characterization of gene expression by this cell type (*Forsthoefel et al., 2012*). More recently, single-cell profiling of whole planarians has distinguished individual intestinal cell types, as well as transitional markers for neoblasts differentiating along endodermal lineages (*Fincher et al., 2018*; *Labbé et al., 2012*; *Zeng et al., 2018*; *Plass et al., 2018*). Both approaches have advanced our understanding of intestinal biology. However, methods that (a) avoid the potentially confounding effects of feeding and dissociation on gene expression and (b) overcome the need to sequence tens of thousands of planarian cells to identify intestinal cells (only 1–3% of all planarian cells, [*Baguñá and Romero, 1981*]), would further enhance the experimental accessibility of the intestine.

Laser-capture microdissection (LCM) was developed as a precise method for obtaining enriched or pure cell populations from tissue samples, including archived biopsy and surgical specimens (*Emmert-Buck et al., 1996*). Since its introduction, LCM has been used to address a vast array of basic and clinical problems requiring genome, transcriptome, or proteome analysis in specific tissues or cell types (*Mahalingam, 2018*; *Bevilacqua and Ducos, 2018*). LCM requires sample preparation including fixation, histological sectioning, and tissue labeling or staining (*Espina et al., 2006*). For subsequent expression profiling, tissue processing must be optimized to maintain morphology and labeling of cells of interest, as well as RNA integrity (*Espina et al., 2006*; *Goldsworthy et al., 1999*; *Gillespie et al., 2002*; *Golubeva and Warner, 2018*). Excision and capture of tissue with infrared and/or ultraviolet lasers is then performed using one of several commercial LCM systems (*Bevilacqua and Ducos, 2018*).

Here, we report the application of LCM for expression profiling of the planarian intestine. We first identified appropriate tissue-processing conditions for extraction of intact RNA from planarian tissue sections. Then, using RNA-Seq, bioinformatics approaches, and whole-mount in situ hybridization, we characterized the intestinal transcriptome. We discovered previously unappreciated regionalization and diversity of intestinal cell types and subtypes, especially amongst goblet cells, and hundreds of intestine-enriched transcripts not identified in recent single-cell profiling efforts. In addition, we identified 22 intestine-enriched transcription factors, including several required for production and/or maintenance of goblet cells, setting the stage for further studies of this cell type. The planarian intestinal transcriptome is a foundational resource for investigating numerous aspects of intestine regeneration and physiology. Furthermore, the LCM methods we introduce offer an additional, complementary strategy for assessing tissue-specific gene expression in planarians.

## Results

### Application of laser-capture microdissection to recover RNA from the planarian intestine

Successful application of laser microdissection requires identification of sample preparation conditions that balance the need to extract high-quality total RNA with preservation of specimen morphology and the ability to identify tissues or cells of interest. Fixation of whole planarians requires an initial step to relax/kill animals and remove mucus, followed by fixation (*Forsthoefel et al., 2014*; *Ross et al., 2015*). We tested three commonly used relaxation/mucus-removal treatments and two

fixatives (formaldehyde and methacarn), separately and together, using short treatment times in order to minimize potential deleterious effects on RNA (*Figure 1—figure supplement 1A*). None of the relaxation treatments detrimentally affected RNA quality, but methacarn (a precipitating fixative) enabled much better RNA recovery than formaldehyde (a cross-linking fixative) (*Figure 1—figure supplement 1A–B*). Next, we assessed how mucus removal affected morphology and staining of cryosections taken from methacarn-fixed planarians, again using a rapid protocol to minimize RNA degradation (*Figure 1—figure supplement 1C*). For all three mucus-removal methods, Eosin Y alone or with Hematoxylin enabled superior demarcation of the intestine, as compared to Hematoxylin alone (*Figure 1—figure supplement 1C*). For preservation of morphology, magnesium relaxation was superior; NAc and HCl treatment caused tearing and detachment of intestine from the slide (*Figure 1—figure supplement 1C*). Finally, we assessed RNA integrity from laser-microdissected intestine and non-intestine from magnesium-treated, methacarn-fixed tissue (*Figure 1A–B* and *Figure 1—figure supplement 1D*), stained only with Eosin Y to further minimize processing time (*Figure 1—figure supplement 1C*). Although additional freezing, cryosectioning, staining, and drying steps required for LCM caused a modest decrease in RNA integrity relative to whole animals (compare *Figure 1—figure supplement 1A* with *1D*), prominent 18S/28S rRNA peaks (which co-migrate in planarians, as in some other invertebrates [*Ishikawa, 1977*; *Matz, 2002*; *Winnebeck et al., 2010*; *Asai et al., 2015*; *Figure 1—figure supplement 1D*]) indicated that the combination of magnesium-induced relaxation, brief methacarn fixation, and rapid Eosin Y staining were suitable for LCM and extraction of RNA of sufficient quality for RNA-Seq.

## Identification of intestine-enriched transcripts and mediolateral expression domains

Using our optimized conditions, we laser microdissected intestinal and non-intestinal tissue from four individual planarians (biological replicates) (*Figure 1A–B* and *Figure 1—figure supplement 1D*). For this study, we isolated tissue from the anterior of the animal (rostral to the pharynx, planarians' centrally located feeding organ), where intestinal tissue is more abundant. We microdissected tissue from medial and lateral intestine separately, since the intestine ramifies into secondary, tertiary, and quaternary branches along the mediolateral axis, but whether gene expression varies along this axis has not been addressed systematically. We then extracted total RNA, conducted RNA-Seq, and identified transcripts that were preferentially expressed in intestinal vs. non-intestinal tissue (*Figure 1C–F* and *Supplementary files 1* and *2*).

Altogether, we detected 13,136 of 28,069 transcripts in the reference transcriptome (46.8% coverage) in non-intestine, medial intestine, and/or lateral intestine (*Figure 1C*). Of these, 1844 were upregulated in either medial or lateral intestine, or both (*Figure 1C*). Specifically, in medial intestine, 1748 transcripts were significantly upregulated (fold-change >2 and FDR-adjusted p-value<0.01) compared to non-intestine (*Figure 1D*). In lateral intestine, 1627 transcripts were upregulated (*Figure 1D*). Although most (1,531/1,844) transcripts were upregulated in both medial and lateral intestine, a small subset of transcripts was significantly upregulated *only* in medial (217/1,844) or lateral (96/1,844) intestine (*Figure 1D*), relative to non-intestine. To further define medial and lateral transcript enrichment, we calculated a 'mediolateral ratio' of the medial and lateral intestine/non-intestine fold-changes (*Figure 1E* and *Supplementary file 1*). Although most transcripts were only modestly enriched (<1.5X fold-change enrichment) in medial (1,124) or lateral (567) intestine, a small number of transcripts was >1.5X enriched in medial (97) or lateral (56) intestine (*Figure 1E* and *Supplementary file 1*).

We validated RNA-Seq results using whole-mount in situ hybridization (WISH) to test expression in fixed, uninjured planarians (*Umesono et al., 1997*; *Pearson et al., 2009*; *King and Newmark, 2013*; *Figure 1F* and *Figure 1—figure supplement 2*). 143/162 transcripts (~88%) had detectable expression in the intestine (*Figure 1—figure supplement 2* and *Supplementary file 1*). Most transcripts were expressed uniformly throughout the intestine (e.g. *ATPase H+-transporting accessory protein 2* (*atp6ap2*), *cytochrome P450 2B19* (*cyp2b19*), *cytochrome P450 2A6* (*cyp2a6*), and *prosaposin* (*psap*); *Figure 1F*, blue borders, and *Figure 1—figure supplement 2*). By contrast, transcripts predicted by RNA-Seq to be most enriched (>1.5X) in medial intestine branches (e.g. *carboxypeptidase A2* (*cpa2*), *rapunzel 4* (*rpz4*), and *gastric triacylglycerol lipase* (*lipf*), green borders, *Figure 1F*) or lateral intestine branches (e.g. a *carboxypeptidase* homolog (*ct14378*), *serine peptidase inhibitor Kunitz type 3* (*spint3*), and a novel gene ('*novel*'), orange borders, *Figure 1F*) were indeed expressed

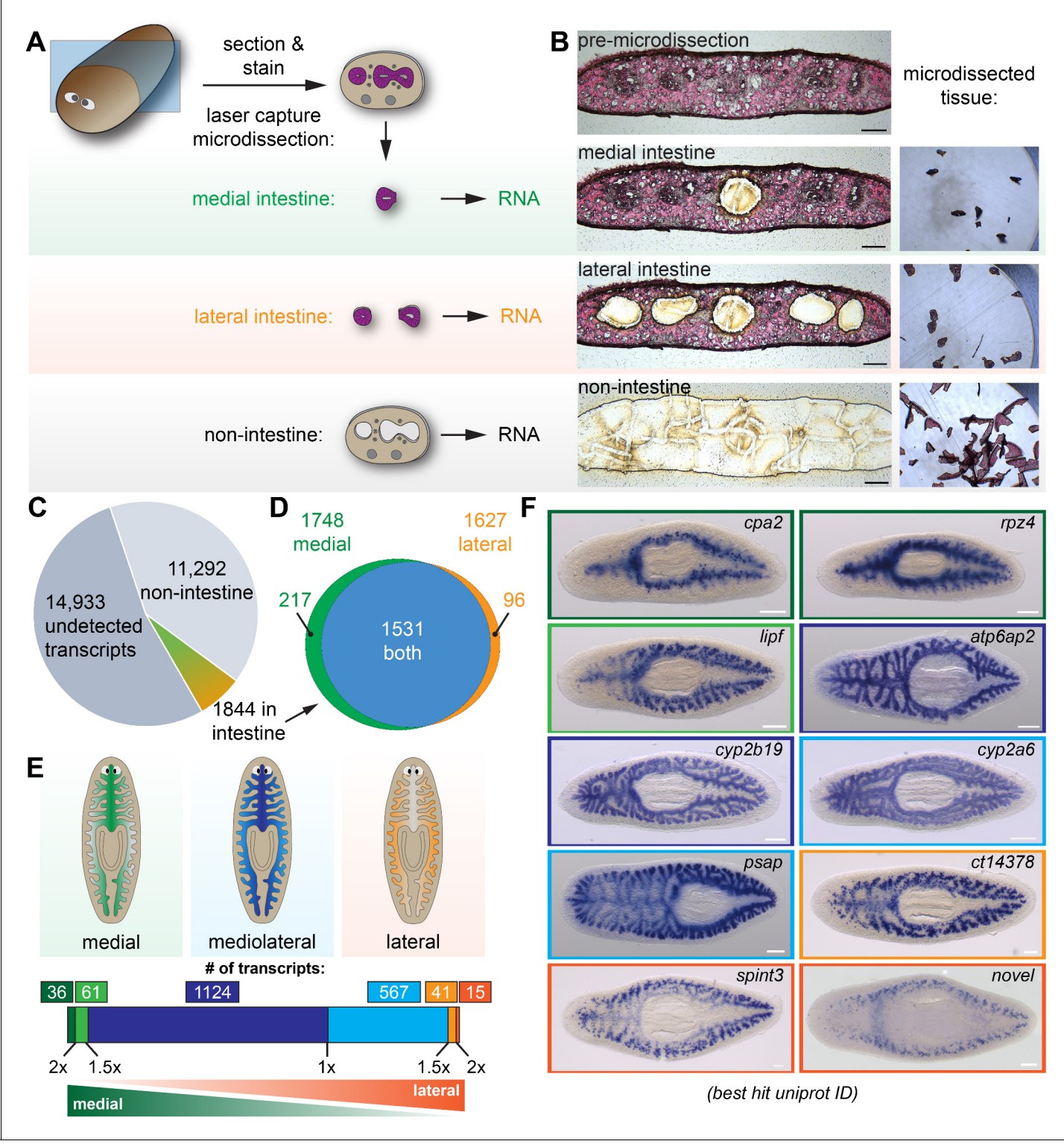

**Figure 1.** Laser-capture microdissection coupled with RNA-Seq identifies 1844 intestine-enriched transcripts. (**A**) Schematic of microdissection workflow. Planarians were fixed and cryosectioned, then sections were stained with Eosin Y. Medial intestine, lateral intestine, and non-intestine tissue were then laser captured, followed by RNA extraction and RNA-Seq. (**B**) Images of an eosin-stained section as tissue is progressively removed (left) and captured (right), yielding three samples with medial intestine, lateral intestine, and non-intestinal tissue. (**C**) Pie chart of RNA-Seq results: of 28,069 total transcripts, 13,136 were detected. Of these, 1844 were upregulated significantly in either medial or lateral intestine. (**D**) Venn diagram showing overlap of medially and laterally enriched intestinal transcripts. (**E**) Schematic of the number of transcripts with enrichment in medial or lateral intestine,

*Figure 1 continued on next page*

*Figure 1 continued*

expressed as a ratio of Fold Changes (FC) in each region. Dark green, FC-medial/FC-lateral > 2 x. Green, FC-medial/FC-lateral = 1.5x-2x. Dark blue, FC-medial/FC-lateral = 1x-1.5x. Blue, FC-lateral/FC-medial = 1x-1.5x. Orange, FC-lateral/FC-medial = 1.5x-2x. Dark orange, FC-lateral/FC-medial > 2 x. (F) Examples of transcripts expressed in the intestine (WISH) in medial (top), mediolateral (middle), and lateral (bottom) regions. Color outlines correspond to the color bar in panel F. Detailed numerical data and gene ID information are available in *Supplementary file 1* and in Results. Scale bars, 100 μm (B), 200 μm (F).

The online version of this article includes the following figure supplement(s) for figure 1:

**Figure supplement 1.** Optimization of fixation and histological staining for laser microdissection.

**Figure supplement 2.** Expression of transcripts enriched in laser-captured intestinal tissue.

at higher levels in these regions. Additionally, although we did not explicitly compare anterior and posterior gene expression, we also discovered anteriorly (a *C-type lectin (Zgc:171670/dd_79)*, *Figure 1—figure supplement 2*) and posteriorly (*lysosomal acid lipase (lipa/dd_122)*, *Figure 1—figure supplement 2*) enriched transcripts, consistent with the influence of anteroposterior polarity cues on intestinal morphology (*Gurley et al., 2008*; *Petersen and Reddien, 2008*; *Iglesias et al., 2008*; *Reuter et al., 2015*; *Thi-Kim Vu et al., 2015*; *Stückemann et al., 2017*). In summary, using LCM together with RNA-Seq identified >1800 intestine-enriched transcripts, and revealed previously unappreciated regional gene expression domains in the intestine.

## Identification of genes expressed by three distinct intestinal cell types

Previously, we identified genes preferentially expressed by intestinal phagocytes (*Forsthoefel et al., 2012*). To distinguish transcripts expressed by phagocytes and other intestinal cell types, such as goblet cells (*Ishii, 1965*; *Garcia-Corrales and Gamo, 1986*; *Garcia-Corrales and Gamo, 1988*), we directly compared $\log_2$ fold-change values for 1317 transcripts represented in our sorted phagocyte data (*Forsthoefel et al., 2012*) as well as laser-microdissected intestinal tissue (this study) (*Figures 2A, C and E*, and *Figure 2—source data 1*). 900/1,317 transcripts were significantly upregulated in both phagocytes and laser-microdissected intestine (*Figure 2A*). We analyzed expression of 82 of these transcripts using WISH (*Figure 2B*, *Supplementary file 1*, and *Figure 1—figure supplement 2*). As expected, the majority (74/82) of these transcripts, which included previously identified intestinal markers *hnf-4* and *nkx2.2* (*Wagner et al., 2011*; *Forsthoefel et al., 2012*; *Garcia-Fernàndez et al., 1993*), displayed uniform, ubiquitous expression throughout the intestine, consistent with enrichment in phagocytes (*Figure 2B* and *Figure 1—figure supplement 2*).

358 intestine-enriched transcripts were not significantly up- or down-regulated in phagocytes (*Figure 2C*, *Figure 1—figure supplement 2*, and *Supplementary file 1*). WISH analysis suggested these transcripts are enriched in multiple intestinal cell types (*Figure 2D*). Some transcripts in this group were expressed ubiquitously throughout the intestine (*ral guanine nucleotide dissociation stimulator-like 1 (rgl1)* and *family with sequence similarity 21 member C (fam21c)*, a homolog of *WASH complex subunit 2*, *Figure 2D*), suggesting expression in phagocytes, possibly in addition to other cell types. However, others were expressed in a distinct subset of less abundant intestinal cells (*peptidase inhibitor 16 (pi16)* and *serine peptidase inhibitor, Kunitz type 3 (spint3)*, *Figure 2D*). These transcripts are enriched in goblet cells, since their WISH expression pattern is highly similar to labeling of this subpopulation by lectins (*Zayas et al., 2010*), antibodies (*Ross et al., 2015*; *Reuter et al., 2015*; *Bueno et al., 1997*), and other recently identified transcripts (*Fincher et al., 2018*; *Plass et al., 2018*; *Reuter et al., 2015*). A third set of transcripts was enriched in basal regions of the intestine (*zgc:172053*, a homolog of human C-type lectin *collectin-10*, and *calmodulin-3 (calm3)*, *Figure 2D*). This pattern resembles that of a planarian *gli*-family transcription factor (*Rink et al., 2009*) and several solute carrier-family transporters (*Thi-Kim Vu et al., 2015*), and indicates expression by 'outer intestinal cells' (which we refer to as 'basal cells' because of their proximity to the basal region of phagocytes) that were also recently identified in a large-scale, single-cell sequencing effort (*Fincher et al., 2018*).

Finally, 59 intestine-enriched transcripts were significantly downregulated in phagocytes (*Figure 2E*, *Figure 1—figure supplement 2*, and *Supplementary file 1*). As expected, we never observed uniform, phagocyte-like expression patterns for these transcripts (*Figure 2F* and *Figure 1—figure supplement 2*). Rather, transcripts in this group were enriched only in goblet cells

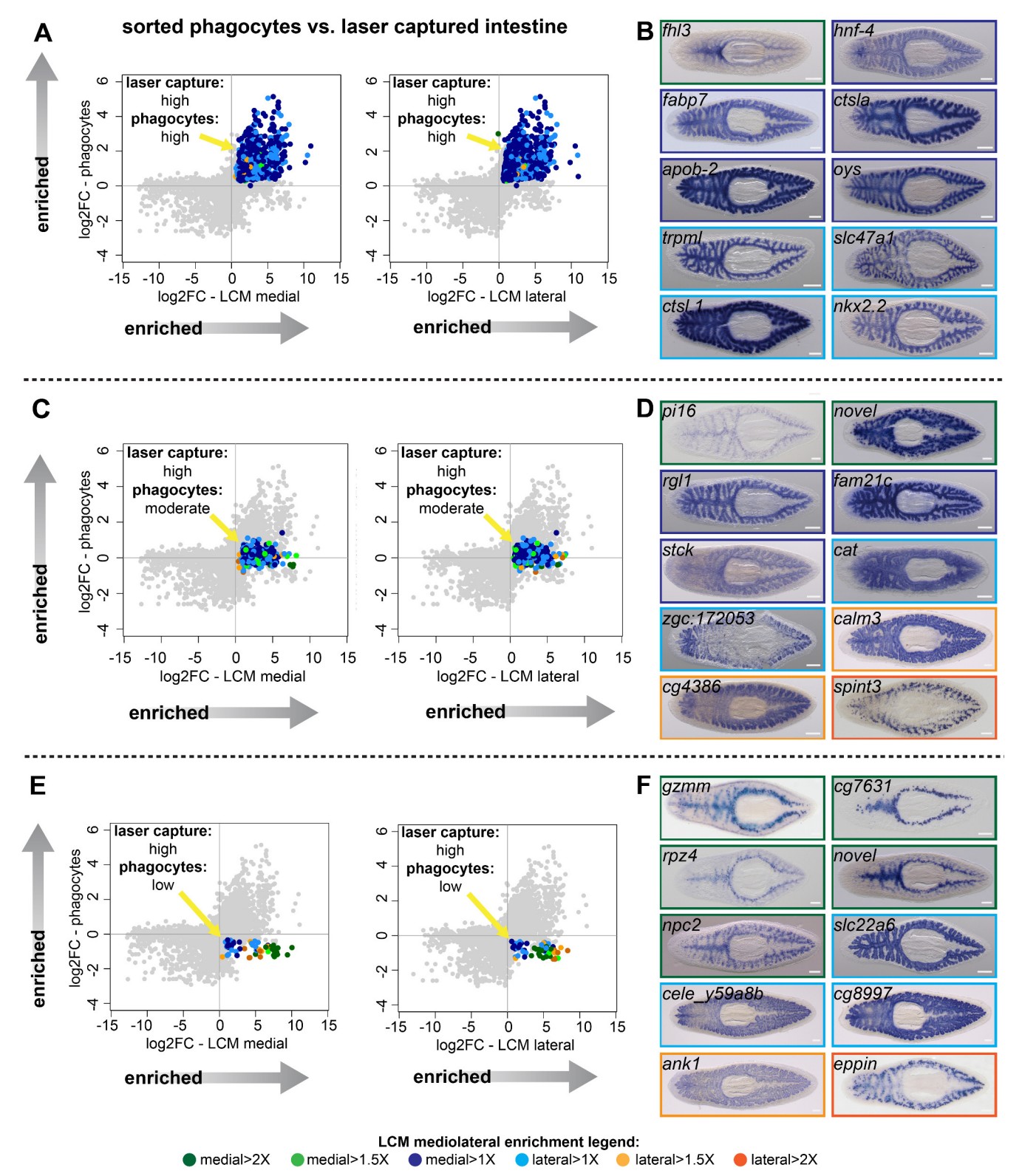

**Figure 2.** Identification of transcripts enriched in specific intestine cell types and regions. (A) Log$_2$ fold-changes for laser-microdissected medial (left) and lateral (right) intestinal tissue (relative to non-intestinal tissue) are plotted on the x-axis, while log$_2$ fold-changes for sorted/purified intestinal phagocytes (compared to all other cell types) are plotted on the y-axis. Transcripts in the upper right quadrant (colorized according to the legend in *Figure 1* and at the bottom of this figure) are expressed preferentially in laser-captured intestine (fold-change >2, FDR-adjusted p value<0.01) *and*

*Figure 2 continued on next page*

*Figure 2 continued*

preferentially in sorted phagocytes (fold-change >2, FDR-adjusted p value<0.05) (phagocytes: 'high'). Most of these transcripts are not medially or laterally enriched in LCM transcriptomes. (B) Whole-mount in situ hybridizations on uninjured planarians showing examples of expression patterns for transcripts in (A). Borders are colorized according to the mediolateral legend in *Figure 1* and at the bottom of the figure. Expression patterns are mostly uniform and ubiquitous in the intestine, consistent with phagocyte-specific expression, with the exception of *fhl3* (top left), which is medially enriched. (C) Plots as in (A), but with colorized transcripts expressed preferentially in laser-captured intestine, but not significantly up- or down-regulated in sorted phagocytes (FDR-adjusted p value>0.05) (phagocytes: 'moderate'). Some of these transcripts are medially or laterally enriched in LCM transcriptomes. (D) Examples of gene expression for transcripts in (C). A variety of intestine expression patterns is observed. (E) Plots as in (A), with transcripts in the lower right quadrant enriched in laser-microdissected intestine, but significantly downregulated (fold-change <2, FDR-adjusted p value<0.05) in sorted phagocytes relative to non-phagocytes (phagocytes: 'low'). Many of these transcripts are enriched in medial or lateral LCM transcriptomes. (F) Examples of gene expression patterns for transcripts in (E). A majority of these transcripts are enriched in goblet or basal cells, sometimes in medial or lateral subpopulations. Detailed gene ID information and numerical data are available in *Supplementary file 1*, *Figure 2— source data 1*, and in Results. Scale bars, 200 μm.

The online version of this article includes the following source data and figure supplement(s) for figure 2:

**Source data 1.** Comparison of transcripts enriched in laser-captured intestine and sorted intestinal phagocytes.
**Figure supplement 1.** Validation of cell-type specific mRNA expression by WISH and correlation with single-cell analysis.

(e.g. *epididymal secretory protein E1/Niemann-Pick disease type C2 protein (npc2)*) or basal cells (e.g. *solute carrier family 22 member 6 (slc22a6)*). Furthermore, some goblet-cell-specific transcripts also appeared to be either medially (e.g. *metalloendopeptidase (cg7631)*) or laterally (e.g., *eppin*) enriched (*Figure 2F* and *Figure 1—figure supplement 2*), suggesting possible specialization of this cell type along the mediolateral axis.

Overall, validation by WISH identified 91 transcripts with a ubiquitous intestinal expression pattern; nearly all of these were upregulated in our sorted phagocyte data (*Figure 1—figure supplement 2* and *Figure 2—figure supplement 1A*). By contrast, most of the 25 validated goblet-cell-enriched transcripts (*Figure 2—figure supplement 1B*) and 25 basal-cell-enriched transcripts (*Figure 2—figure supplement 1C*) were not upregulated in sorted phagocytes. We have made all WISH expression patterns and RNA-Seq data available in an interactive website, https://plangut.omrf.org.

## Multiple cell types and novel subtypes reside in the planarian intestine

To further characterize intestinal cell types and subtypes, we used fluorescence in situ hybridization (FISH) to investigate co-expression of intestine-enriched transcripts. First, we verified the existence of three distinct cell types, using highly expressed phagocyte, goblet, and basal-specific markers (*Figure 3A–C*). Expression of the most phagocyte-enriched transcript, *cathepsin La (ctsla)*, was ubiquitous throughout intestinal branches, but did not overlap with *npc2*, a goblet-cell-enriched mRNA (*Figure 3A*), or with *slc22a6*, a basally enriched transcript (*Figure 3B*). Goblet cell-enriched *npc2* was expressed by cells with minimal *slc22a6* expression (*Figure 3C*), further reinforcing that *npc2+* goblet cells are distinct from *ctsla+* phagocytes as well as *slc22a6+* basal cells. Additional markers validated the distinct identity of these cell types (*Figure 3—figure supplement 1A–B*). We also found that *slc22a6+* basal cells were distinct from visceral muscle fibers that surround intestinal branches, occupying basal regions around digestive cells (*Kobayashi et al., 1998*; *Orii et al., 2002*), consistent with another study (*Scimone et al., 2018*; *Figure 3D–E*). Thus, *slc22a6+* cells represent a novel cell type in the intestine that is distinct from visceral muscles, phagocytes, and goblet cells, and that has, to our knowledge, not been described by numerous previous histological and ultrastructural studies. Our data independently confirm the identification of this cell type in a recent single-cell sequencing effort (*Fincher et al., 2018*).

Using additional markers, we also characterized previously unappreciated heterogeneity in gene expression amongst both goblet and basal cells. These included a subpopulation of goblet cells restricted to medial regions of the intestine, mainly localized to primary branches (*Figure 4A and C*), and a lateral subpopulation within secondary, tertiary, and quaternary branches (*Figure 4B–C*). Only rarely did goblet cells in these medial and lateral domains co-mingle, or co-express both markers at the mediolateral boundaries between primary and secondary branches (*Figure 4C*). We also identified subpopulations of basal cells in lateral regions of the intestine (*Figure 4D* and *Figure 3—figure supplement 1B*). Finally, we also identified transcripts expressed by multiple cell types in different

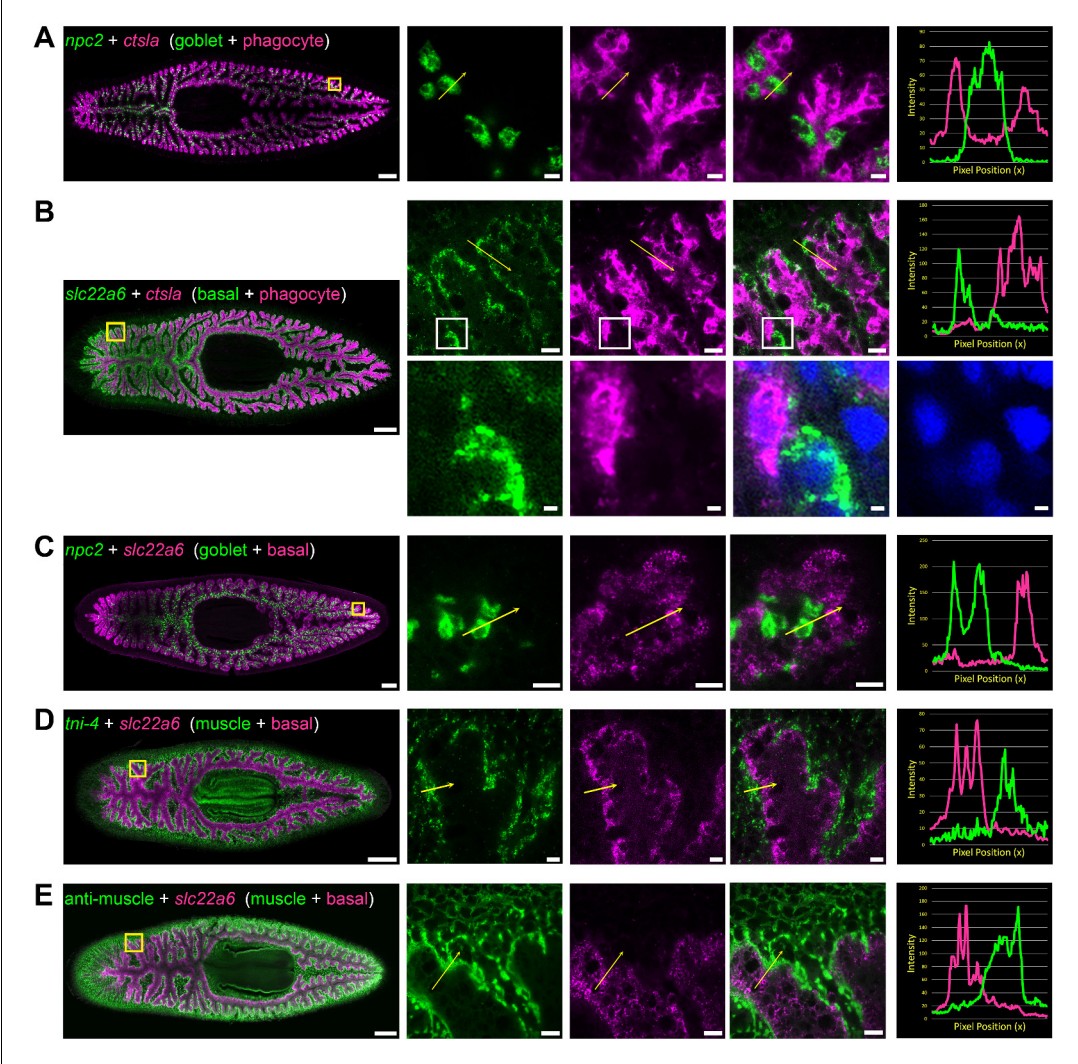

**Figure 3.** Double fluorescence in situ hybridization reveals three major cell types in the planarian intestine. (**A**) Confocal images of *npc2* (green) and *ctsla* (magenta) in situ hybridization. Left to right, whole animal (yellow box indicates magnified region in right panels), zoomed area in green, magenta, and merge (yellow arrow indicates profile line in right-most panel), and a graph showing pixel intensity in each color from tail to head of the yellow profile arrow. *ctsla* is the top phagocyte-specific gene in the phagocyte microarray dataset, while *npc2* is enriched in goblet cells. (**B**) *slc22a6* (green) and *ctsla* (magenta). *slc22a6* mRNA is restricted to the basal region of the intestine, and shows minimal overlap with the phagocyte marker *ctsla*. The white box represents the cropped region shown below with DAPI labeling nuclei, indicating that these riboprobes label distinct cells. (**C**) *npc2* (green) and *slc22a6* (magenta). *npc2* is enriched in goblet cells while *slc22a6* is enriched in basal cells, with minimal overlapping signal. (**D**) *troponin I 4* (*tni-4*, green) (**Witchley et al., 2013**) and *slc22a6* (magenta). *tni-4* is expressed by visceral muscles, while *slc22a6* is found in basal cells. (**E**) Anti-muscle antibody (6G10, green) and *slc22a6* (magenta). Detailed gene ID information is available in **Supplementary file 1** and in Results. Scale bars, whole animals 200 µm; magnified images, 10 µm, magnified crop of basal cell (**B**), 2 µm.

The online version of this article includes the following figure supplement(s) for figure 3:

**Figure supplement 1.** Additional cell-type specific transcripts revealed by double fluorescent in situ hybridization.

combinations and levels (*Figure 4E–G* and *Figure 3—figure supplement 1C–D*), illustrating the complexity of gene expression even in a tissue with relatively few cell types.

## Laser capture substantially increases resolution of the global intestinal transcriptome

We also compared transcript enrichment in phagocytes/enterocytes, goblet cells, and basal cells/ outer intestinal cells reported in three recent single-cell RNA-Seq (scRNA-Seq) analyses of planarian

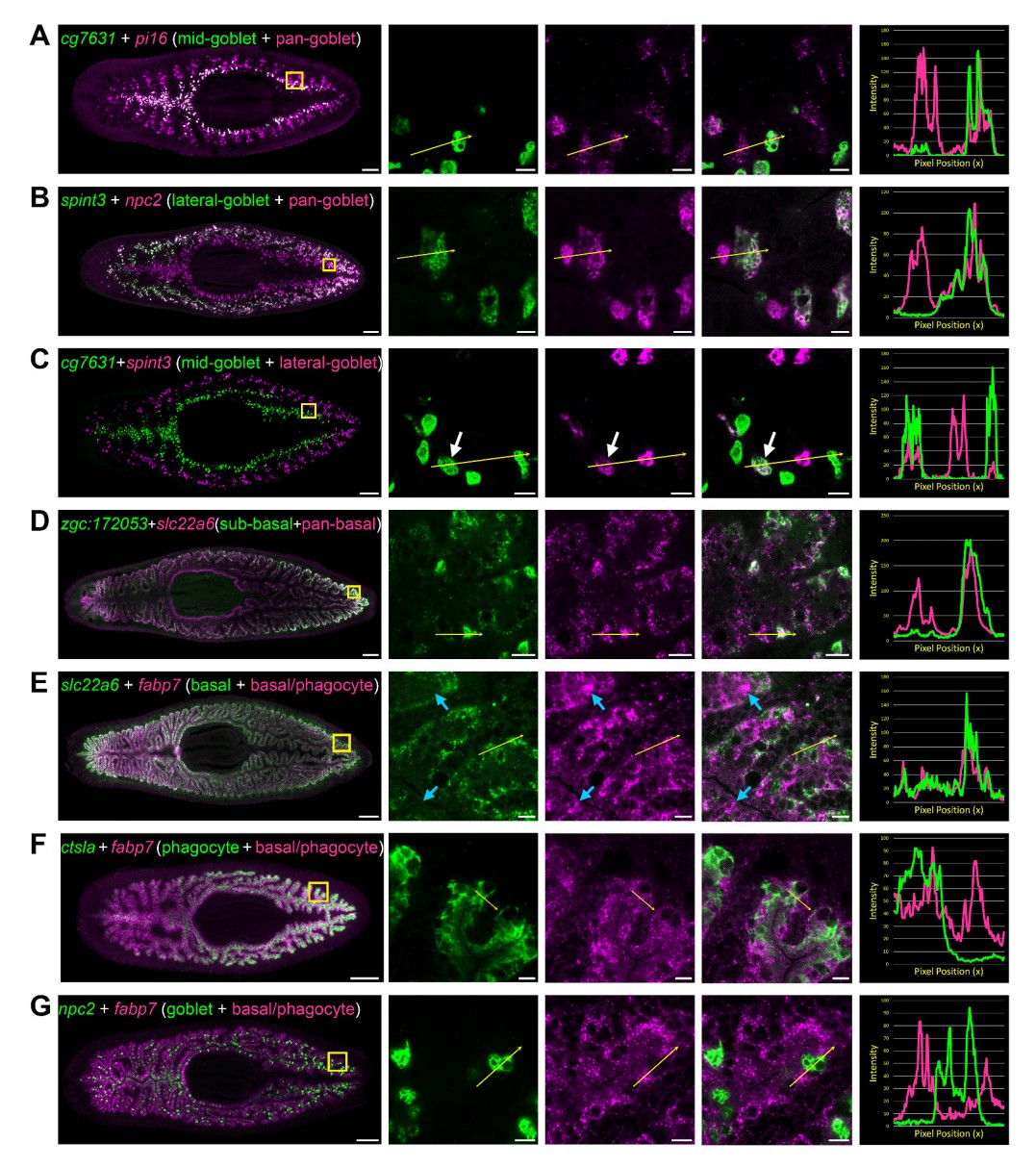

**Figure 4.** Transcripts expressed by intestinal subpopulations and multiple cell types. (**A**) Confocal images of *cg7631* (green) and *pi16* (magenta) in situ hybridization. Left to right, whole animal (yellow box indicates magnified region in right panels), zoomed area in green, magenta, and merge (yellow arrow indicates profile line in right-most panel), and a graph showing pixel intensity in each color from tail to head of the yellow profile arrow. *cg7631* is enriched in medial goblet cells, while *pi16* is found in all goblet cells. (**B**) *spint3* (green) and *npc2* (magenta). *spint3* is enriched in the lateral goblet cell population, while *npc2* is expressed by all goblet cells. (**C**) *cg7631* (green) and *spint3* (magenta). *cg7631* is enriched in medial goblet cells; *spint3* is enriched in lateral goblet cells. Only rarely do these two markers label the same cell, indicated with a white arrow. (**D**) *zgc:172053* (green) and *slc22a6* (magenta). *zgc:172053* is enriched in a subset of basal cells, while *slc22a6* is more ubiquitously enriched in most basal cells. (**E**) *slc22a6* (green) and *fabp7* (magenta). *slc22a6* is a basally enriched gene, while *fabp7* is expressed by both basal cells and more apical cells (phagocytes). Blue arrows indicate apical gene expression where *slc22a6* is absent. (**F**) *ctsla* (green) and *fabp7* (magenta). *ctsla* expression is enriched in phagocytes, while *fabp7* is found in both phagocytes and basal cells. (**G**) *npc2* (green) and *fabp7* (magenta). *npc2* is enriched in goblet cells, and overlaps minimally with *fabp7* in phagocytes and basal cells. Detailed gene ID information is available in *Supplementary file 1* and in Results. Scale bars, whole animals 200 μm; magnified images, 10 μm.

cells (*Fincher et al., 2018*; *Plass et al., 2018*; *Swapna et al., 2018*). There was considerable agreement between our verified in situ expression patterns and cell-type enrichment predicted by scRNA-Seq studies, although we did identify numerous additional cell-type-specific transcripts (*Figure 5A*). In addition, phagocyte-, goblet-, and basal-cell-specific transcripts from scRNA-Seq studies mapped to similar quadrants in our phagocyte vs. laser-captured intestine plots (*Figure 2—figure supplement 1D–H*). Similarly, the majority of phagocyte-enriched transcripts detected in our earlier study (*Forsthoefel et al., 2012*) were also enriched in laser-captured intestine (*Figure 5B*).

Furthermore, we identified 809 intestine-enriched transcripts that single-cell studies did not find to be enriched in the intestine (or for some, any planarian cell type) (*Figure 5C*). Using WISH, we validated intestine enrichment for 22/28 of these mRNAs (*Figure 5D–H*), including transcripts with a uniform/phagocyte-like expression pattern (*Figure 5D*), and others with expression in goblet and basal cells (*Figure 5E–F*). We also note that over 1000 intestine-enriched transcripts in scRNA-Seq studies were not included in our LCM-generated transcriptome (*Figure 5C*). However, the vast majority (>80%) of these were enriched in multiple cell types (*Fincher et al., 2018*; *Plass et al., 2018*; *Supplementary file 2*), suggesting considerable expression in non-intestinal tissue, consistent with our data. The incomplete overlap between various scRNA-Seq studies and our results could be explained, in part, by different log-fold enrichment criteria used to identify cell-type-specific transcripts. However, the detection of transcripts exclusively enriched in laser-captured intestine suggests that expression profiling of laser-captured bulk tissue is more sensitive than current single-cell profiling approaches, and that LCM may be a preferable method for assessing tissue-specific gene expression when single-cell resolution is not required.

## Diverse digestive physiology genes are expressed in the planarian intestine

In order to globally characterize functional classes of genes expressed in the planarian intestine, we assigned Gene Ontology (GO) Biological Process (BP) terms to planarian transcripts based on homology to human, mouse, zebrafish, *Drosophila,* and *C. elegans* genes, and then identified over-represented terms among intestine-enriched transcripts (*Figure 6*, *Figure 6—source data 1A*, and *Supplementary file 3*). Highly represented terms fell broadly into seven groups (*Figure 6*), all of which are related to the intestine's roles in digestion, nutrient storage and distribution, as well as innate immunity. Metabolic processes were among the most highly represented: hundreds of transcripts were predicted to regulate catabolism, biosynthesis, and transport of a variety of macromolecules (e.g. lipids and carbohydrates) and small molecules (e.g. amino acids and ions) (*Supplementary file 3A*). Hundreds of upregulated transcripts were also predicted to regulate molecular transport, vesicular trafficking, and organelle-based import and export (*Figure 6—figure supplement 3A*). These included over 70 members of the solute carrier family of transmembrane transporters (*Supplementary file 3B*), reinforcing the intestine's central role in metabolite transport, and also suggesting a potential role supporting the excretory system in maintaining extracellular solute concentration (*Thi-Kim Vu et al., 2015*; *Andrikou et al., 2019*). Enriched regulators of vesicular trafficking also included nearly 30 Ras-related Rab GTPase proteins (*Supplementary file 3C*). Among regulators of organelle and cellular physiology, transcripts predicted to coordinate phagosome, endosome, and lysosome physiology were among the most highly represented (*Supplementary file 3A*). These included several *vacuolar protein sorting-associated protein (vps)* homologs required for lysosome tethering to late endosomes and autophagosomes (*Spang, 2016*), and homologs of the autophagy-related proteins *atg3* and *atg7*, ubiquitin ligases that are required for autophagosome formation during nutrient starvation (*Komatsu et al., 2005*; *Sou et al., 2008*), and which may contribute to planarians' ability to survive extended fasting (*Felix et al., 2019*). As in our previous study of phagocyte expression (*Forsthoefel et al., 2012*), here we also identified many intestine-enriched genes predicted to regulate cell shape, motility, polarity, and adhesion, including numerous cytoskeletal regulators, regulators of interactions with extracellular matrix, and *partitioning defective 6 (pard6b),* which we previously demonstrated was required for planarian intestinal remodeling (*Forsthoefel et al., 2012*; *Supplementary file 3A*). Finally, transcripts predicted to regulate responses to stress, microorganisms, and other stimuli were also intestine enriched (*Figure 6*). Prominent among these were regulators of innate immunity, including over 30 tumor necrosis factor receptor-associated factor homologs (TRAFs), a family of adaptor proteins that is expanded in *S. mediterranea* (*Swapna et al., 2018*) and function as effectors of receptor signaling in innate and

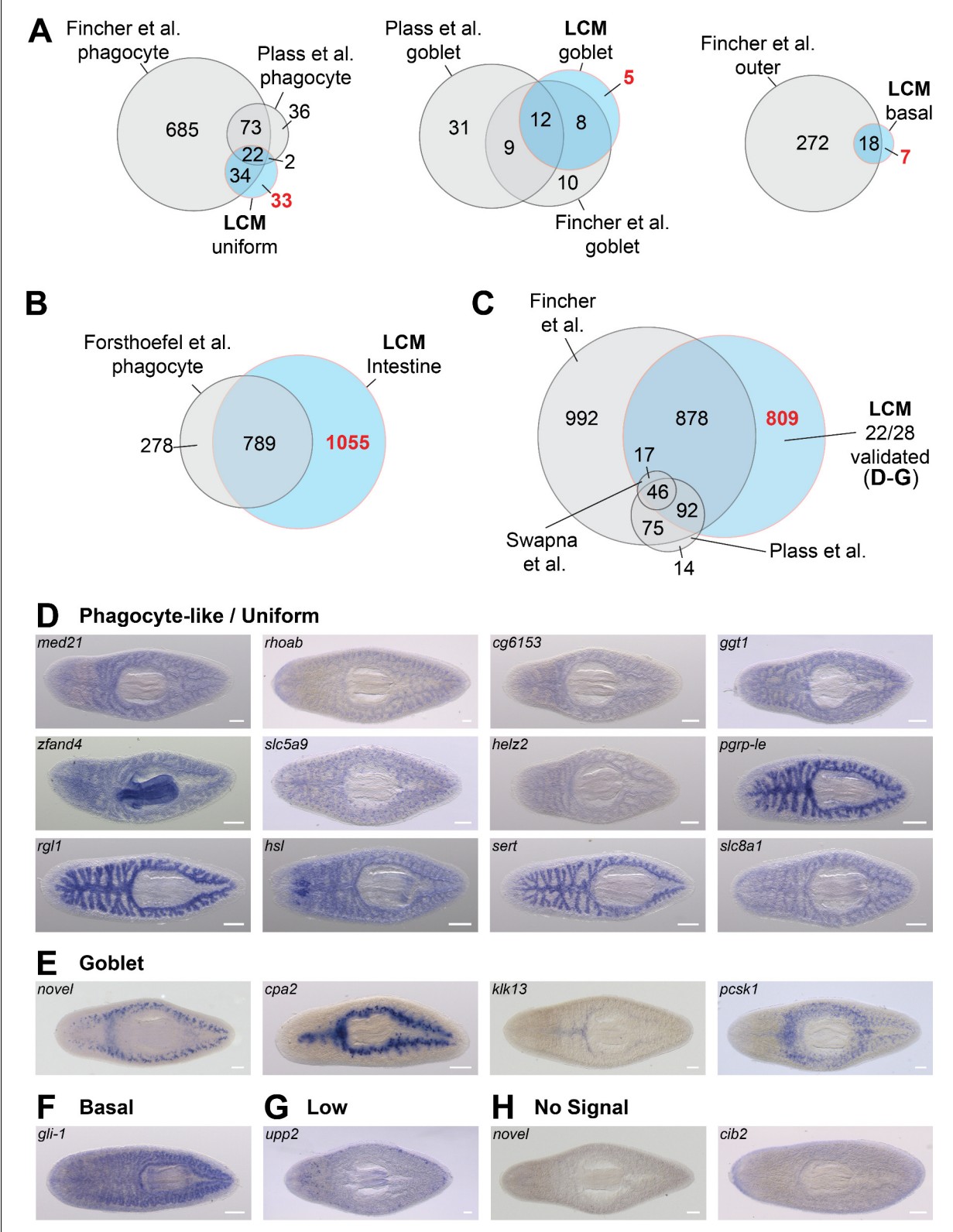

**Figure 5.** LCM-RNA-Seq identifies additional intestine-enriched transcripts. (**A**) Venn diagrams compare intestine-enriched transcripts identified in three scRNA-Seq studies (*Fincher et al., 2018*; *Plass et al., 2018*; *Swapna et al., 2018*) and this study, grouped by assignment to cell type (scRNA-Seq studies), or by uniform/phagocyte-like, goblet cell, and basal cell WISH expression patterns (this study). (**B**) Comparison of transcripts enriched in sorted phagocytes (*Forsthoefel et al., 2012*) with orthologs in the dd_Smed_v6 transcriptome, and laser-captured intestine. (**C**) Overlap between all intestine-

*Figure 5 continued*

enriched transcripts (based on RNA-Seq data) in three recent single-cell studies and our LCM data. Overlaps of two or fewer genes are not displayed. (D) Examples of intestine-enriched transcripts with a uniform/phagocyte-like expression pattern (WISH) identified by LCM-RNA-Seq, but not in other studies. (E) Examples of intestine-enriched transcripts with expression in goblet cells (WISH) identified by LCM-RNA-Seq, but not in other studies. (F) Example of intestine-enriched transcript expressed in basal cells (WISH) identified by LCM-RNA-Seq, but not in other studies. (G) Intestine-enriched transcript with low expression identified in this study. (H) Examples of transcripts enriched in LCM-RNA-Seq data for which expression was undetectable by WISH (e.g. did not validate). Detailed numerical data are in *Supplementary file 1* and *Figure 2—source data 1*. Scale bars, 200 μm.

adaptive immunity (*Xie, 2013*; *Arnold et al., 2016*; *Supplementary file 3D*). Notably, 62 intestine-enriched transcripts (from this study) were previously found to be upregulated in response to shifting planarians from recirculating to static culture conditions, which causes microbiome dysbiosis (*Arnold et al., 2016*; *Figure 6—figure supplement 1A* and *Figure 6—source data 2A*). Further supporting a role for the intestine in innate immunity and/or inflammatory responses, homologs of 99 *S. mediterranea* intestine-enriched transcripts were also upregulated by ingestion of pathogenic bacteria in *Dugesia japonica*, a related planarian species (*Abnave et al., 2014*; *Figure 6—figure supplement 1B* and *Figure 6—source data 2B*).

## Analysis of mediolaterally enriched transcripts reveals potential goblet cell roles

In order to understand whether putative functional roles above are performed by specific intestinal cell types or domains, we also analyzed GO term over-representation among transcripts enriched in scRNA-Seq data (*Fincher et al., 2018*), and in laser-captured medial and lateral intestinal tissue. Most functional categories predicted by LCM transcript analysis (*Figure 6*) were also enriched among phagocyte scRNA-Seq transcripts (*Figure 6—figure supplement 2A* and *Figure 6—source data 1B–D*). Furthermore, GO analysis suggested that basal cells may play a significant role in metabolism and energy processing, and that goblet cells may influence extracellular matrix organization and play specialized roles in lipid metabolism (*Figure 6—figure supplement 2A* and *Figure 6—source data 1B–D*).

Analysis of biological process GO term enrichment among all 1221 medial and 623 lateral transcripts – without regard to cell-type specificity – identified numerous putative regulators of innate immunity and macromolecular catabolism among medial transcripts, and of extracellular matrix organization among lateral transcripts (*Figure 6—figure supplement 2B–D* and *Figure 6—source data 1E–H*). Intriguingly, many transcripts with the greatest medial (97) or lateral (56) enrichment (e. g. >1.5X in medial vs. lateral tissue or vice versa) in the intestine were expressed by goblet cells (*Figure 2C–F*, *Supplementary file 1*, and *Figure 1—figure supplement 2*). GO analysis of these transcripts suggested possible functional specialization with respect to lipid metabolism, protein processing, extracellular matrix organization, and innate immunity (*Figure 6—figure supplement 2E* and *Figure 6—source data 1I–J*).

Analysis of expression in situ supported these predictions (*Figure 6—figure supplement 3A–D*). For example, in support of previous suggestions that goblet cells promote luminal digestion (*Arnold, 1909*; *Pedersen, 1961*; *Jennings, 1962*), we identified several goblet-enriched transcripts predicted to encode secreted regulators of protein catabolism (*pancreatic carboxypeptidase A2*) and triglyceride catabolism (*lipase F/gastric triacylglycerol lipase*) (*Figure 6—figure supplement 3A*). Second, we also identified three goblet-enriched *kallikreins* (*klk*) (*Figure 6—figure supplement 3A* and *Figure 1—figure supplement 2*), secreted proteases whose mammalian homologs produce vasoactive plasma kinin, but also hydrolyze extracellular matrix molecules, growth factors, hormone proteins, and antimicrobial peptides in numerous tissues (*Prassas et al., 2015*). Kallikreins are expressed by goblet cells in rat, cat, and mouse intestines (*Schachter et al., 1986*; *Grün et al., 2016*), and have been implicated in inflammatory bowel disease and gastrointestinal cancers (*Stadnicki, 2011*; *Kontos et al., 2013*), suggesting additional conservation of planarian goblet cell physiology. Third, goblet cells express *peptidoglycan recognition protein (pgrp-1b)* (*Figure 6—figure supplement 3A*), whose vertebrate and invertebrate homologs modulate innate immune signaling and play direct bactericidal roles (*Kurata, 2014*; *Dziarski and Gupta, 2018*), suggesting goblet cells may coordinate immune responses and/or regulate microbiome composition. Consistent with this idea, a second planarian paralog, *pgrp-1e* (*Figure 1—figure supplement 2*), is upregulated in the

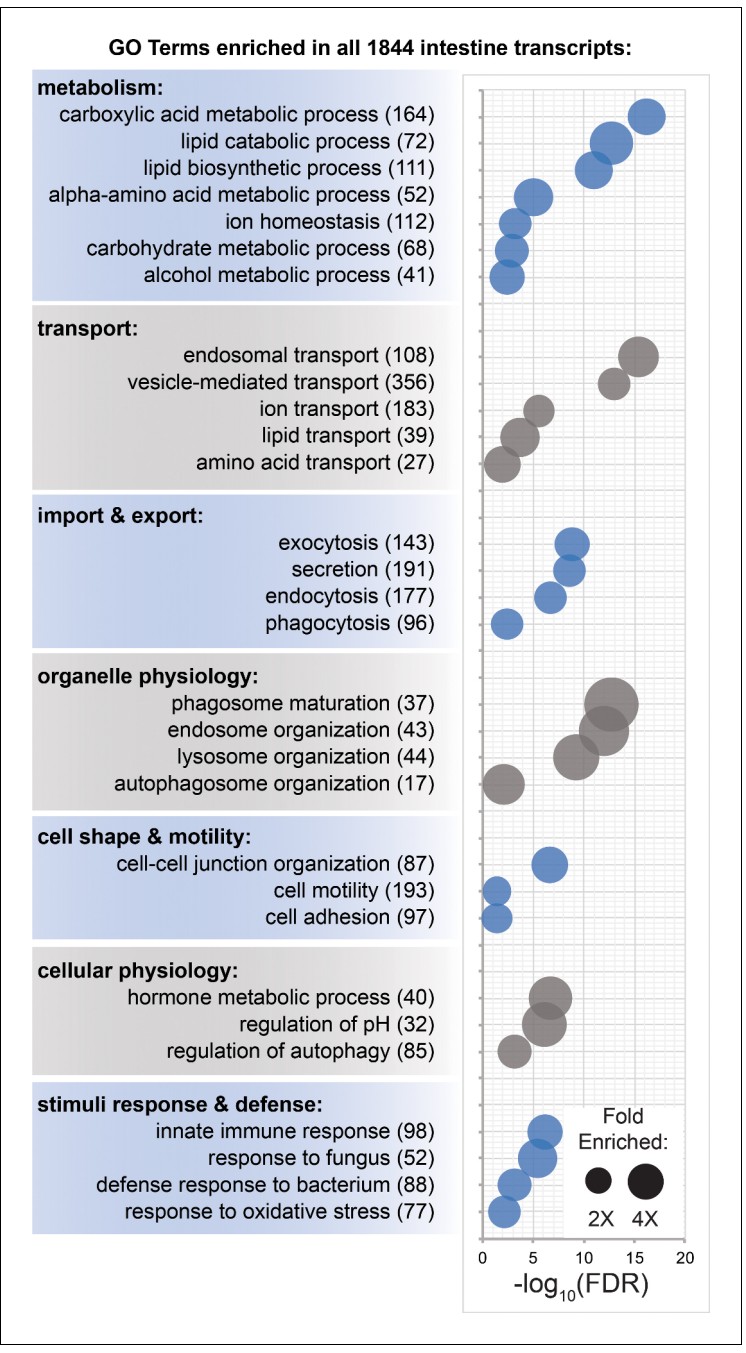

**Figure 6.** Transcripts involved in metabolism, transport, organelle physiology, and stimuli responses are enriched in the intestine. (**A**) Biological process Gene Ontology terms significantly over-represented in intestine-enriched transcripts. Bubble size indicates fold enrichment relative to all transcripts detected in laser-captured tissue, while position on the x-axis indicates FDR-adjusted significance. Numbers in parentheses indicate the number of intestine-enriched transcripts annotated with each term. Detailed numerical data are in *Figure 6—source data 1*. The online version of this article includes the following source data and figure supplement(s) for figure 6:

**Source data 1.** Gene Ontology biological process term enrichment for intestine-enriched transcripts.
**Source data 2.** LCM intestine-enriched transcripts in innate immunity studies.
**Figure supplement 1.** Innate immunity-related transcripts enriched in the intestine.
**Figure supplement 2.** Enriched biological process GO terms for transcripts enriched in intestinal cell types and regions.
**Figure supplement 3.** Gene expression by goblet cells suggests multiple physiological roles.

intestine (but not restricted to goblet cells) in response to *Pseudomonas* infection (*Arnold et al., 2016*). Fourth, a homolog of *prohormone convertase (pcsk1)* is enriched in goblet cells, as well as peripharyngeal cells surrounding the pharynx (*Figure 6—figure supplement 3C*). Prohormone convertases (PCs) cleave neuropeptide and peptide hormone preproteins to generate bioactive peptides. In both vertebrates and invertebrates, PCs function in neurons, but also in endocrine cell types such as pancreatic islet cells and digestive tract enteroendocrine cells, where they regulate glucose levels, energy homeostasis, and appetite (*Grün et al., 2016*; *Pauls et al., 2014*; *Stijnen et al., 2016*). Although another planarian *pcsk* paralog, *Smed-pc2,* processes neuropeptides required for germline development (*Collins et al., 2010*), to our knowledge, no intestine-enriched prohormones have been reported. Nonetheless, *pcsk1* expression suggests goblet cells may play an enteroendocrine-like role, as in other organisms. Finally, we were surprised to find that genes encoding planarian homologs of gel-forming mucins (which we identified in separate bioinformatic searches) were expressed not by goblet cells, but by peripharyngeal cells that send projections into the pharynx (*Forsthoefel et al., 2014*; *Figure 6—figure supplement 3D*). While additional goblet-enriched proteins with mucin-like roles might exist (*Bocchinfuso et al., 2012*; *Syed et al., 2008*), these expression patterns suggest that some planarian gel-forming mucin proteins are delivered to the intestinal lumen through the pharynx, and indicate a possible difference between planarian and vertebrate intestinal goblet cells (*Birchenough et al., 2015*; *Chang et al., 1994*; *Johansson et al., 2011*).

## Evolutionary conservation of human digestive organ gene expression

Further illustrating the conservation of physiological roles, we found that the intestine expresses numerous homologs of transcripts that are enriched in human GI tissues (*Figure 7A–E*, *Supplementary file 4*, and *Figure 7—source data 1*). To make this comparison, we conducted reciprocal best homology (RBH) searches (*Bork et al., 1998*; *Tatusov et al., 1997*) to identify 5583 planarian transcripts encoding predicted open reading frames (ORFs) with high homology to ORFs in UniProt human transcripts (*The UniProt Consortium, 2018*), and vice versa (*Figure 7A–B*). Of these, we then identified 5561 transcripts (including 699 gut-enriched transcripts) with human transcripts represented in the Human Protein Atlas (*Figure 7B–C*), in which transcripts whose unique or highly enriched tissue-specific expression defines 32 human tissues or organs (*Uhlén et al., 2015*). Next, we calculated the percentage of all (5,561) and gut-enriched (699) RBH transcripts that were enriched in human tissues (*Figure 7D*), then expressed these percentages as a fold-enrichment ratio (planarian intestine vs. non-intestine) to estimate similarity between the planarian intestine transcriptome and the transcriptomes of human tissues (*Figure 7E*).

Strikingly, four of the five tissues to which the planarian intestine was most similar are involved in digestion (duodenum, small intestine, and gallbladder) or energy storage/metabolism (liver) (*Figure 7E*). The intestine was also similar to other human digestive tissues (esophagus, colon, and rectum), as well as kidney, possibly suggesting a role supporting the planarian protonephridial system in filtration or processing of extracellular solutes (*Figure 7E*; *Thi-Kim Vu et al., 2015*; *Rink et al., 2011*; *Scimone et al., 2011*). Duodenum- and small intestine-enriched RBH transcripts included predicted regulators of endodermal specification, bile transport, lipid metabolism, and glucose transport (*Supplementary file 4*). In addition, RBH homologs enriched in liver included several predicted regulators of glucose, amino acid, lipid, and xenobiotic compound metabolism (*Supplementary file 4*). These observations reinforce the evolutionary conservation of intestinal gene expression and indicate that physiological roles performed by multiple human digestive organs are consolidated in the planarian intestine.

## Intestine-enriched transcription factors regulate goblet cell differentiation and maintenance

Definition of the intestinal transcriptome enables identification of genes required for regeneration and functions of distinct intestinal cell types. Here, to initiate this effort, we focused on goblet cells, which expressed the majority of medially and laterally enriched transcripts we identified (*Supplementary file 1*, *Figure 1—figure supplement 2*). Ultrastructurally, planarian goblet cells possess numerous large proteinaceous granules and abundant rough endoplasmic reticulum (*Willier et al., 1925*; *Ishii, 1965*; *Garcia-Corrales and Gamo, 1986*; *Pascolini and Gargiulo, 1975*), resembling mammalian goblet cells that produce a protective mucous barrier and mount innate

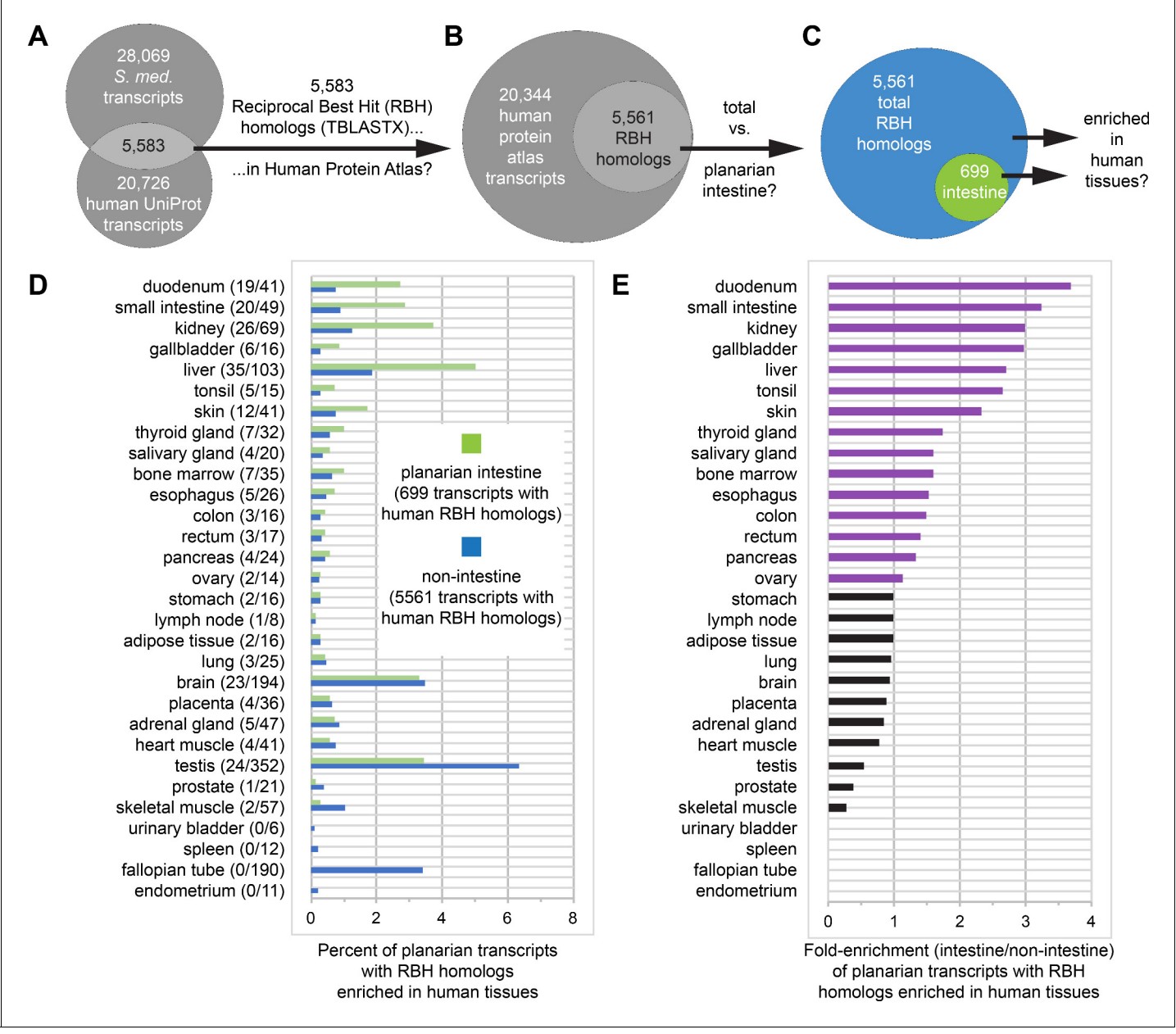

**Figure 7.** Homologs of planarian intestinal transcripts are enriched in human digestive tissues. (**A**) 5583 *S. mediterranea* and *Homo sapiens* UniProt transcripts hit each other in reciprocal TBLASTX queries. (**B**) 5561/5583 human UniProt RBH homologs of planarian transcripts were present in the Human Protein Atlas. (**C**) 699/5561 RBH homologs were enriched in the planarian intestine. (**D**) Enrichment of RBH homologs in human tissues. The first number in parentheses is the number of planarian intestine-enriched RBH homologs (of 699) and the second number in parentheses is the number of all RBH homologs (of 5561) for each human tissue. Histogram bars represent percentage of planarian transcripts with RBH homologs enriched in human tissues. For example, 19/699 (2.72%) UniProt RBH homologs of planarian intestine-enriched transcripts were tissue-enriched, tissue-enhanced, or group-enriched in the duodenum, while only 41/5561 (0.74%) of all UniProt RBH homologs of planarian transcripts were similarly enriched. (**E**) Fold enrichment (planarian intestine/non-intestine) of RBH homologs for each human tissue. Histogram bars were calculated as a ratio of percentages in panel D. For example, 2.72% of 699 planarian intestinal RBH homologs and 0.74% of all 5561 planarian RBH homologs were enriched in the duodenum, yielding fold enrichment of 2.72/0.74 = 3.68X. Detailed numerical data are available in *Figure 7—source data 1*.

The online version of this article includes the following source data for figure 7:

**Source data 1.** Planarian transcripts with reciprocal best hit (RBH) homologs enriched in human tissues.

immune responses (*Birchenough et al., 2015*; *Dalton, 1952*; *Freeman, 1966*; *McCauley and Guasch, 2015*; *Knoop and Newberry, 2018*). However, although numerous markers and reagents have been identified that label planarian goblet cells (*Fincher et al., 2018*; *Plass et al., 2018*; *Ross et al., 2015*; *Reuter et al., 2015*; *Zayas et al., 2010*; *Bueno et al., 1997*), to our knowledge, genes required for goblet cell differentiation, maintenance, or physiological roles have not been reported. Initially, we used RNAi to assess the roles of 16 of the most medial and 8 of the most lateral goblet-enriched transcripts (*Supplementary file 5A-B*). However, we did not observe failure to feed, decreased viability, or defects in blastema formation, even after 8 weeks of knockdown (feeding 1x/week) (*Supplementary file 5A-B*). This might be due to functional redundancy, since multiple lipases, carboxypeptidases, and kallikriens are expressed by goblet cells. Alternatively, other intestinal cell types might play overlapping roles with respect to some functions.

Reasoning that transcription factors (TFs) would regulate goblet cell generation and/or maintenance, we next focused on transcripts encoding 22 intestine-enriched TFs, only 10 of which were previously known to be enriched in intestinal cells (*Supplementary file 5C*). Using FISH, we validated expression of all but one of these TFs in the intestine (*Figure 8—figure supplement 1*). In a dsRNA-mediated RNA interference screen to specifically assess goblet cell regeneration, we found that knockdown of three TFs dramatically reduced expression of a goblet cell marker in regenerating head, trunk, and tail fragments (*Figure 8A–B*, *Figure 8—figure supplement 2*, and *Supplementary file 5C*). Knockdown of *mediator of RNA polymerase II transcription subunit 21 (med 21)* reduced goblet cells and blastema formation (*Supplementary file 5C*), but also caused severe disruption of intestinal integrity in our previous study (*Forsthoefel et al., 2012*). This suggested a non-goblet-cell-specific role, and we did not investigate *med21* further. Knockdown of a second transcription factor, *gli-1* (a transducer of hedgehog signaling [*Rink et al., 2009*; *Glazer et al., 2010*]), caused failure of goblet cells to regenerate at the midline of the new anterior branch in amputated tail fragments regenerating a new head (*Figure 8A–B*, *Supplementary file 5C*, and *Figure 8—figure supplement 2A*). In addition, goblet cells were less abundant in pre-existing regions of the intestine, particularly in lateral intestinal branches (*Figure 8A–B*, *Figure 8—figure supplement 2A–C*). Regeneration of goblet cells was also reduced in new tail branches of *gli-1 (RNAi)* head fragments (*Figure 8—figure supplement 2B*) and in anterior and posterior branches in *gli-1(RNAi)* trunk fragments (*Figure 8—figure supplement 2C*). These effects on goblet cells were specific, since phagocytes (*Figure 8A*, *Figure 8—figure supplement 2A–C*) and basal cells (*Figure 8B*) regenerated normally. Although we infrequently observed smaller posterior blastemas characteristic of reduced hedgehog signaling (*Rink et al., 2009*; *Glazer et al., 2010*; *Yazawa et al., 2009*), phagocytes regenerated normally in this region (*Figure 8—figure supplement 2C*).

By contrast to the *gli-1* phenotype, goblet cells appeared to differentiate normally in regenerating intestine upon knockdown of a third TF, *ras-responsive element binding protein 2 (RREB2)*, including at the midline of anterior intestinal branches in tail fragments (*Figure 8A–B*; *Figure 8—figure supplement 2A*), in posterior branches in head fragments (*Figure 8—figure supplement 2B*), and in both anterior and posterior branches in trunk fragments (*Figure 8—figure supplement 2C*). However, in tail and trunk regenerates, new goblet cells were largely restricted to the midline/primary branches (*Figure 8A–B*, *Figure 8—figure supplement 2A and C*), and were less abundant in posterior branches of head fragments (*Figure 8—figure supplement 2B*). Furthermore, in pre-existing intestinal regions (especially lateral branches), goblet cell numbers were dramatically reduced or even completely absent. These included the posterior of tail fragments (*Figure 8A–B*; *Figure 8—figure supplement 2A*), the anterior of head fragments (*Figure 8—figure supplement 2B*), and central regions of trunk fragments (*Figure 8—figure supplement 2C*). As with *gli-1*, phagocytes and basal cells were unaffected (*Figure 8A–B*; *Figure 8—figure supplement 2A–C*). Together, these results suggest that *gli-1* regulates neoblast fate specification and/or differentiation of neoblast progeny into goblet cells in new intestinal branches, while *RREB2* may control maintenance or survival of goblet cells after they initially differentiate. For both knockdowns, the reduction of goblet cells in lateral and pre-existing primary branches could be a consequence of reduced differentiation (*gli-1*) or maintenance/survival (*RREB2*) in these regions prior to amputation, during regeneration, or both.

In uninjured animals, both *gli-1* and *RREB2* also reduced goblet cell numbers. Knockdown of *gli-1* reduced goblet cell numbers in uninjured animals, especially in anterior, posterior, and lateral intestine branches (*Figure 8C*), but many goblet cells remained in medial, primary intestinal branches. In *RREB2(RNAi)* animals, goblet cells were more dramatically reduced in all regions of the intestine

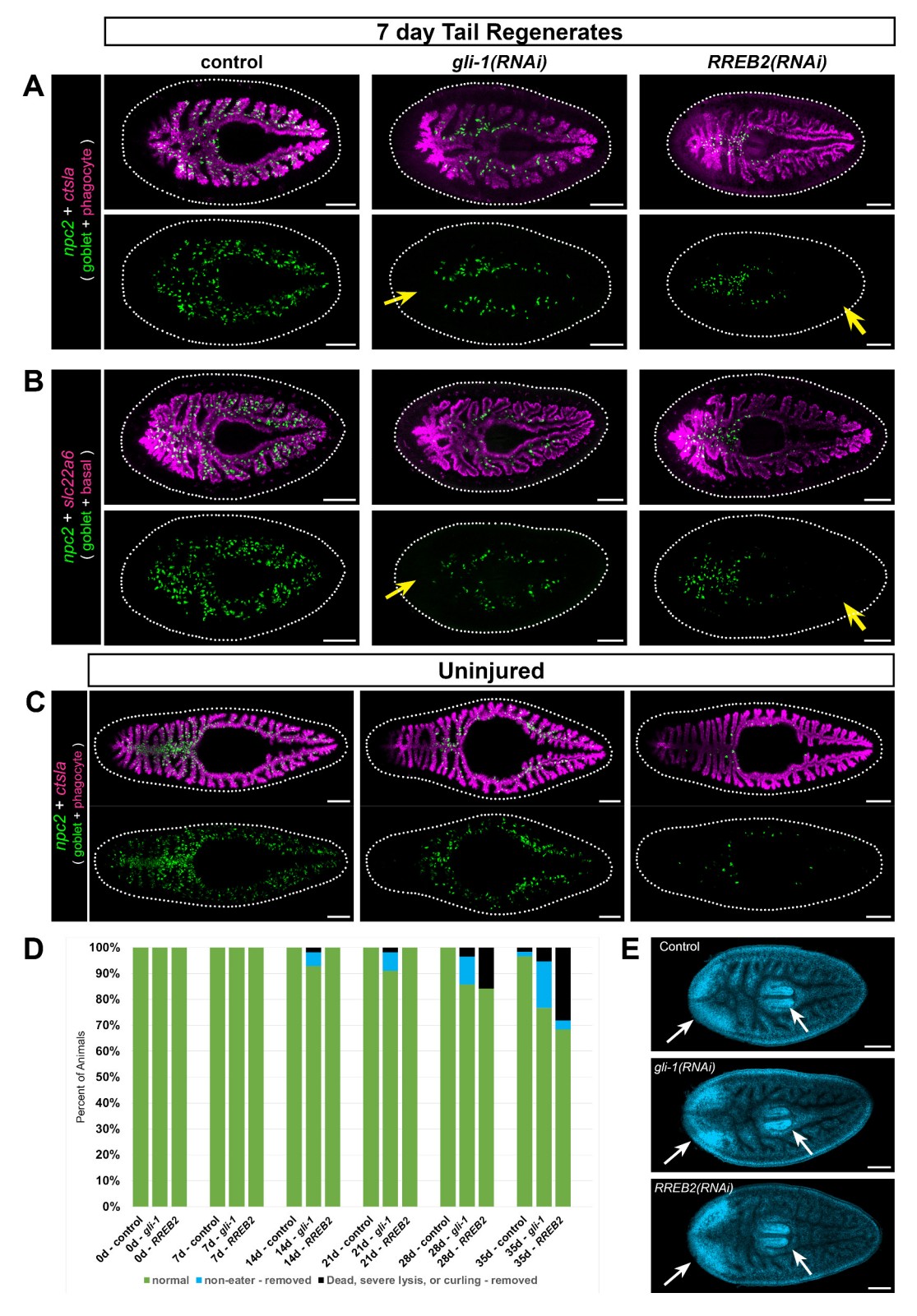

**Figure 8.** *gli-1* and *RREB2* regulate goblet cell abundance. (**A**) In 7 day tail regenerates, *gli-1* knockdown dramatically reduces goblet cells (*npc2+*) at the midline in regenerating intestine (yellow arrow), while *RREB2* knockdown reduces goblet cells in old tissue (yellow arrow). Phagocytes (*ctsla+*) appear normal in all conditions. Both *gli-1(RNAi)* and *RREB2(RNAi)* also reduce goblet cells in lateral branches. (**B**) Basal cells (*slc22a+*) are unaffected in *gli-1(RNAi)* and *RREB2(RNAi)* regenerates, while goblet cells are reduced similar to A. (**C**) In uninjured animals, *gli-1* RNAi causes moderate goblet cell

*Figure 8 continued on next page*

*Figure 8 continued*

loss, while *RREB2* RNAi results in severe goblet cell loss. (D) Phenotypes in *gli-1(RNAi)* and *RREB2(RNAi)* planarians during six dsRNA feedings (once per week). Animals refuse food and undergo lysis and death with increasing frequency over the RNAi time course. Total sample size was n ≥ 55 for each condition; data were pooled from three independent biological replicates of n ≥ 18 each. (E) DAPI labeling of tail regenerates shown in B. White arrows indicate normal regeneration of new brain and pharynx. Animals in A, B, and E were fed dsRNA eight times (twice per week), starved 7 days, amputated, then fixed 7 days later. Animals in C and D were fed dsRNA six times (once per week), starved 7 days, then fixed for FISH. Detailed gene ID and RNAi phenotype information is available in *Supplementary file 1*, *Supplementary file 5*, and *Figure 8—source data 1A*. Scale bars, 200 µm.

The online version of this article includes the following source data and figure supplement(s) for figure 8:

Source data 1. (F8SD1-A) Detailed feeding and viability numerical data for three biological replicates conducted for control, *gli-1*, and *RREB2* RNAi phenotypes; (F8SD-2) individual measurements for area and length of control, *gli-1*, and *RREB2* knockdowns before any treatment and after five dsRNA feedings; (F8SD-3) statistics tables for length and area measurements.

Figure supplement 1. Expression of intestine-enriched transcription factors.
Figure supplement 2. Goblet cells are reduced in *gli-1* and *RREB2* 7 day regenerates.
Figure supplement 3. Effects of *gli-1* and *RREB2* knockdown on medial and lateral goblet cell subpopulations.
Figure supplement 4. Expression patterns of *gli-1* and *RREB2*.
Figure supplement 5. Additional genes whose knockdown affects goblet cell numbers in 7 day regenerates.
Figure supplement 6. Phenotypes, area, and length of *gli-1* and *RREB2* knockdowns.

(*Figure 8C*). Although these phenotypes were broadly consistent with observations in regenerates, they also suggested that *gli-1* and *RREB2* might be required for differentiation and/or maintenance of distinct mediolateral goblet cell subpopulations in uninjured animals. Indeed, we found (using additional goblet cell markers) that almost no lateral goblet cells remained in *gli-1(RNAi)* uninjured animals, while again, medial goblet cells were much less affected (*Figure 8—figure supplement 3A*). By contrast, in *RREB2(RNAi)* uninjured animals, reduction of both medial and lateral goblet cell numbers was pronounced, but nonetheless some lateral goblet cells remained (*Figure 8—figure supplement 3A*). These differential effects on mediolateral subpopulations were not observed in regenerates. For example, both subpopulations failed to differentiate in the primary anterior intestinal branch in *gli-1(RNAi)* tail regenerates (*Figure 8—figure supplement 3B*), and both subpopulations were severely reduced in pre-existing, posterior branches of *RREB2(RNAi)* tail regenerates (*Figure 8—figure supplement 3B*).

Taken together, these results suggest that, in uninjured animals undergoing normal homeostatic growth and renewal, *gli-1* is primarily required for differentiation of new lateral intestinal cells. Alternatively, as goblet cells fail to renew, remaining goblet cells might somehow migrate to more medial intestinal branches, or survival of goblet cells in primary branches could be prolonged. Conversely, *RREB2* seems to be required for maintenance of both medial and lateral goblet cells, although lateral cell numbers are slightly less affected by *RREB2* knockdown, compared to *gli-1*. Finally, in regenerates, *gli-1* and *RREB2* knockdown affects medial and lateral goblet cells similarly (but in regenerating vs. pre-existing intestine). These complex observations suggest that mechanisms regulating goblet cell differentiation and maintenance may differ during homeostasis and regeneration, or that compensatory mechanisms capable of sustaining goblet cell production/survival in uninjured *gli-1(RNAi)* and *RREB2(RNAi)* animals are insufficient to meet the increased demand for new tissue during regeneration.

Despite their specific effects on goblet cells, neither *gli-1* nor *RREB2* mRNAs are specifically enriched in this cell type. In fact, *gli-1,* which was previously shown to be expressed by intestine-associated cells (*Rink et al., 2009*; *Currie et al., 2016*), was most highly enriched in phagocytes, basal cells, and intestine-associated muscle cells (visceral muscle), but was also expressed at lower levels in some goblet cells (*Figure 8—figure supplement 4A*). *RREB2* was enriched in goblet cells and basal cells, but expression was also observed in some phagocytes and intestine-associated muscle (*Figure 8—figure supplement 4B*). Intriguingly, we found that knockdown of two other TFs, *48 related 1 (fer1)* (also called *pancreas transcription factor one subunit alpha* or *PTF1A* [*Fincher et al., 2018*]), and *LIM homeobox 2 (lhx2b)*, also modestly reduced goblet cells in pre-existing branches (*Figure 8—figure supplement 5A–C* and *Supplementary file 5*). mRNAs encoding these TFs are enriched in basal regions of the intestine (*Figure 6—figure supplement 1*). In addition, single-cell transcriptome data suggest *PTF1A* expression is elevated in differentiating neoblast progeny in the basal cell lineage (*Fincher et al., 2018*), and *PTF1A* also reduces basal intestinal cell numbers

(*Fincher et al., 2018*). Thus, although our data support a role for *gli-1* and *RREB2* in goblet cells and/or their precursors, they also raise the possibility that basal cells (and possibly phagocytes or muscle cells) may non-autonomously influence goblet cell differentiation and/or survival.

## Goblet cell reduction compromises feeding behavior and viability

We asked whether goblet cell depletion affected planarian viability, behavior, or regeneration. Over a 6 week dsRNA feeding regimen, some uninjured *gli-1(RNAi)* and *RREB2(RNAi)* planarians refused food and failed to eat after 4–5 weeks (*Figure 8D*). In both *gli-1(RNAi)* and *RREB2(RNAi)* animals, some animals eventually lysed, curled, or died (*Figure 8—figure supplement 6A*), suggesting that goblet cells are required for viability. Additionally, we observed a modest (but insignificant) decrease in animal size (*Figure 8—figure supplement 6B–C*) in *RREB2(RNAi)* planarians relative to controls. Interestingly, we only observed feeding failure and other phenotypes in animals fed the dsRNA/liver mix 1x/week (every 7 days) for 6 weeks; animals that were fed 2x/week (every 3–4 days), but still six times, did not refuse to eat, lyse, curl, or die (not shown). This suggests that goblet cells may regulate hunger (or other aspects of digestive physiology) primarily in starved animals, consistent with expression of *prohormone convertase* (*Figure 6—figure supplement 3C*), above. Next, to assess regeneration, we amputated planarians fed 2x/week for eight feedings, in order to eliminate the influence of feeding failure on possible regeneration phenotypes. In both *gli-1(RNAi)* and *RREB2 (RNAi)* regenerates, goblet cell depletion was robust (*Figure 8A–B*), but neither gene was required more generally for regeneration (with the infrequent exception of reduced posterior blastemas in *gli-1(RNAi)* regenerates, mentioned above), as the brain (*Figure 8E*), pharynx (*Figure 8E*), other intestinal cell types (*Figure 8A–B*), and new intestinal branches (*Figure 8A–B*) all regenerated without noticeable defects. Together, these results suggest that goblet cells are broadly dispensable for regeneration, and that their primary role may be to regulate appetite or other aspects of intestinal physiology that contribute to viability. Alternatively, it is possible that functions in other cell types may underlie the *gli-1* and *RREB2* feeding and viability phenotypes. In addition, some goblet cell functions may be required only when planarians are challenged by stresses like bacterial infection or extended starvation, possibilities that will require further investigation.

## Discussion

We have developed methods for applying laser-capture microdissection to planarian tissue, which we used to define gene expression and cell types in the intestine (*Figure 9A–C*). The intestine expresses genes involved in metabolism, nutrient storage and transport, innate immunity, and other physiological roles, demonstrating considerable functional homology with digestive systems of other animals, including humans. Comparison of gene expression in microdissected tissue to that of intestinal phagocytes (previously isolated by sorting) enabled identification of transcripts enriched in two other cell types: goblet cells and basal cells. We also discovered previously unappreciated intestinal cell-type diversity, especially amongst goblet cells, which reside in distinct medial and lateral intestinal domains. Identification of medially and laterally enriched transcripts suggests an additional paradigm for addressing axial influences on organ regeneration, as well as evolution of patterning mechanisms influencing the regionalization of bilaterian digestive systems (*Martindale and Hejnol, 2009*; *Telford et al., 2015*). Finally, we identified intestine-enriched transcription factors that play distinct roles in goblet cell differentiation and maintenance, and found that depletion of goblet cells reduces planarians' willingness to feed and viability, with only negligible effects on regeneration of other intestinal cell types or non-intestinal tissues.

Characterization of the planarian intestinal transcriptome provides a framework for several next steps. First, a detailed description of intestinal cell types will facilitate further studies of digestive cell types. Here, we identified two transcription factors, *gli-1* and *RREB2,* whose RNAi phenotypes suggest that distinct mechanisms govern differentiation and maintenance of goblet cells. We are unaware of studies in other organisms that have uncovered direct roles for either *gli-1* or *RREB2* in goblet cells. However, in mice, modulation of Sonic Hedgehog (an upstream activator of Gli-family TFs) levels affects goblet cell numbers (*Gagné-Sansfaçon et al., 2014*; *Liang et al., 2012*), and RREB1 cooperatively regulates expression of the peptide hormone secretin in enteroendocrine cells (*Ray et al., 2003*). By contrast, in the *Drosophila* midgut, Hedgehog (Hh) signaling promotes intestinal stem cell proliferation (*Tian et al., 2015*), while the RREB-1 ortholog *hindsight/pebbled*

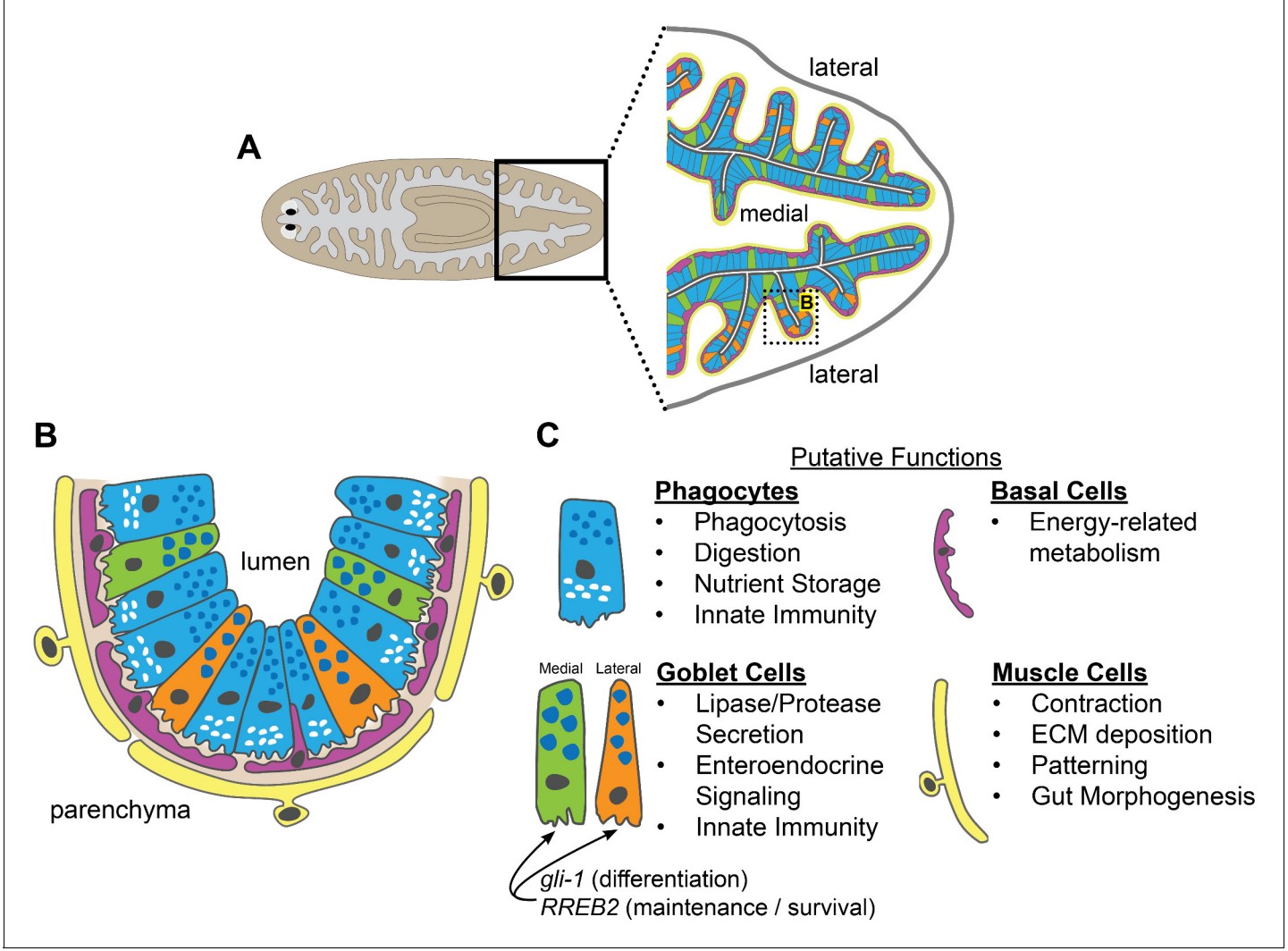

**Figure 9.** Schematic of intestinal cell types and putative functions. (**A**) Illustration of cell types and locations in intestinal branches in the planarian tail. Phagocytes (blue); medial goblet cells (green); lateral goblet cells (orange); basal cells (magenta); visceral muscle (yellow). (**B**) Magnified view/horizontal section of the boxed area in (**A**), showing cell types and locations in one intestinal branch. Cell type colors as in (**A**). Basement membrane (light brown). (**C**) Putative cell type functions inferred from Gene Ontology, cell-type specific transcript expression, and published studies. Intestinal muscle functions inferred from *Scimone et al. (2018)*.

suppresses midgut intestinal stem cell proliferation and is required for differentiation of absorptive enterocytes (*Baechler et al., 2016*). In planarians, hedgehog (Hh) regulates anteroposterior polarity, neoblast proliferation, and neurogenesis (*Rink et al., 2009*; *Yazawa et al., 2009*; *Currie et al., 2016*; *Wang et al., 2016*); our data suggest a possible additional role for Hh in intestinal morphogenesis. Similarly, although a second planarian RREB paralog, *RREBP1,* is partially required for eye regeneration and may promote differentiation (*Mihaylova et al., 2018*), implication of *RREB2* in goblet cell maintenance/survival suggests that Ras-mediated signaling could also influence cellular composition in the intestine (although direct links to Ras remain to be established). Thus, although characterization of the precise roles of *gli-1* and *RREB2* will be required, our results suggest broad similarity between planarians and other animals with respect to evolutionarily conserved signaling pathways that govern cell dynamics in the digestive tract. Furthermore, although we focused on goblet cells, previous studies suggest that several genes (*gata4/5/6-1, hnf4, egfr-1, PTF1A*) expressed by cycling neoblast subpopulations (*Barberán et al., 2016*; *Wagner et al., 2011*; *van Wolfswinkel et al., 2014*; *Zeng et al., 2018*) may be required for differentiation of multiple intestinal cell types (*Barberán et al., 2016*; *Fincher et al., 2018*; *Flores et al., 2016*; *González-*

*Sastre et al., 2017*). Thus, enumeration of intestinal cell types and subtypes here, along with endo-derm-specific progenitors in several recent single-cell analyses (*Fincher et al., 2018*; *Zeng et al., 2018*; *Plass et al., 2018*), provides a rich list of candidate regulators for further elucidation of differentiation, maintenance, and functions of planarian digestive cells and their progenitors.

Second, functional analysis of intestine-enriched transcripts will help to resolve the intestine's role in regulation of the stem cell microenvironment. Knockdowns of the intestine-enriched TFs *nkx2.2* or *gata4/5/6-1* result in reduced proliferation and blastema production (*Forsthoefel et al., 2011*; *Flores et al., 2016*). In addition, a subset of *tetraspanin group-specific gene-1 (tgs-1)*-positive neo-blasts likely to be pluripotent neoblasts resides near intestinal branches (*Zeng et al., 2018*), further hinting at a niche-like role for the intestine. Numerous intestine-enriched transcripts encode regulators of metabolite processing and transport, as well as putatively secreted proteins, suggesting multiple possible cell non-autonomous influences on neoblast dynamics. Because gut-enriched TFs like *nkx2.2* and *gata4/5/6-1* are also expressed by neoblasts and other cell types (*Fincher et al., 2018*), LCM also provides an efficient approach for clarifying which candidate stem cell regulators are dependent on these or other TFs for their intestine-specific expression.

Third, LCM provides a complementary method for assessing injury-induced gene expression changes in the intestine and other planarian tissues that may overcome the shortcomings of other available methods. Previously, we developed a method for purification of intestinal phagocytes from planarians that ingested magnetic beads (*Forsthoefel et al., 2012*; *Forsthoefel et al., 2014*). Laser microdissection may be preferable, because it enables detection of transcripts expressed by other intestinal cell types (not just phagocytes), and also eliminates potentially confounding gene expression changes caused by feeding. Similarly, although single-cell sequencing (SCS) approaches in planarians have driven significant advances in our understanding of planarian cell types and their responses to injury (*Fincher et al., 2018*; *van Wolfswinkel et al., 2014*; *Zeng et al., 2018*; *Plass et al., 2018*; *Wurtzel et al., 2015*; *Wurtzel et al., 2017*), LCM may provide a more direct and efficient way to assess gene regulation in tissues with rarer cell types. For example, because intestinal cells comprise only 1–3% of total planarian cells (*Baguñá and Romero, 1981*), without prior enrichment SCS would potentially require analysis of tens of thousands of cells to reliably detect gene expression changes in the intestine. In addition, although chemistry and computational approaches for SCS are improving rapidly (*Wu et al., 2017*; *Soneson and Robinson, 2018*), LCM may enable more sensitive detection of low-copy transcripts or subtle fold changes in bulk tissue, and also bypass 'noise' caused by dissociation or other SCS-related technical artefacts.

Regeneration of digestive organs is not well understood. Development of a robust method for isolating intestinal tissue, and characterization of the intestinal transcriptome, will facilitate mechanistic studies in planarians. To facilitate exploration of the planarian intestinal transcriptome as a resource, we have developed a website, plangut.omrf.org. The methods and resources presented here will also support comparative analyses, complementing current and future efforts to understand digestive tract regeneration in platyhelminths, sea cucumbers, annelids, ascidians, amphibians, and mammals (*Goodchild, 1956*; *O'Steen, 1958*; *Takeo et al., 2008*; *Kaneko et al., 2010*; *Zattara and Bely, 2011*; *Okano et al., 2015*; *Ortiz-Pineda et al., 2009*; *Sun et al., 2011*; *Zhang et al., 2017*; *Ayyaz et al., 2019*). In addition, studies in planarians and other regeneration models are likely to generate new insights into cellular processes (e.g. proliferation, differentiation, metabolism, and stress responses) whose dysregulation underlies human gastrointestinal pathologies associated with aging and disease.

## Materials and methods

**Key resources table**

| Reagent type (species) or resource | Designation | Source or reference | Identifiers | Additional information |
|---|---|---|---|---|
| Strain, strain background (*Schmidtea mediterranea*) | Asexual clonal line CIW4 of *Schmidtea mediterranea* | PMID:12421706 | RRID: NCBITaxon:79327 | All animals used in this study |

*Continued on next page*

*Continued*

| Reagent type (species) or resource | Designation | Source or reference | Identifiers | Additional information |
|---|---|---|---|---|
| Recombinant DNA reagent | pBluescript II SK(+) (plasmid) | Agilent Technologies | Cat:212205 | For cloning from ESTs |
| Recombinant DNA reagent | pJC53.2 (plasmid) | PMID:20967238 | RRID: Addgene_26536 | For cloning |
| Antibody | (mouse, monoclonal) Muscle antibody 6G10 | doi: 10.1186/s12861-014-0050-9 | | Used at 1:2000 |
| Chemical compound, drug | Formaldehyde | EMD Millipore | Cat:FX0410-5 | Used at 4% in 1xPBS |
| Chemical compound, drug | Platinium Taq | Invitrogen | Cat:10966026 | For PCR |
| Chemical compound, drug | Trizol | Invitrogen | Cat:15596026 | Used for RNA extraction |
| Chemical compound, drug | RNAseZAP | Invitrogen | Cat:AM9780 | For LCM |
| Chemical compound, drug | Mayer's Hematoxylin | Sigma Aldrich | Cat:MHS16-500ML | For LCM |
| Chemical compound, drug | Alcoholic Eosin Y | Sigma Aldrich | Cat:HT110116-500ML | For LCM |
| Commercial assay or kit | RNA Screen Tape | Agilent | Cat:5067–5576 | Used to verify RNA quality |
| Commercial assay or kit | PicoPure RNA Isolation Kit | Arcturus | Cat:12204–1 | For LCM |
| Commercial assay or kit | Quantseq 3' mRNA Library Prep Kit FWD | Lexogen | Cat:K01596 | For RNA-seq |
| Commercial assay or kit | iScript Kit | Bio-Rad | Cat:1708891 | For cDNA synth |
| other | PEN membrane slides | Leica | Cat:11505158 | For LCM |
| Software, algorithm | Bestus Bioinformaticus Duk | DOE Joint Genome Institute | RRID:SCR_016969 | RNAseq read trimming |
| Software, algorithm | FastQC | Babraham Institute | RRID:SCR_014583 | RNAseq quality check |
| Software, algorithm | Bowtie2 | DOI: 10.1038/nmeth.1923 | | RNAseq transcript mapping |
| Software, algorithm | Samtools v1.3 | PMID:19505943 | RRID:SCR_002105 | RNAseq processing |
| Software, algorithm | edgeR v3.8.6 | PMID:19910308 | RRID:SCR_012802 | RNAseq differential expression |
| Software, algorithm | TBLASTX | U.S. National Library of Medicine | RRID:SCR_011823 | For Human gene comparison |
| Software, algorithm | BLASTX | U.S. National Library of Medicine | RRID:SCR_001653 | Homology searches |
| Software, algorithm | BiNGO | PMID:15972284 | RRID:SCR_005736 | Gene Ontology |
| Software, algorithm | NCBI ORFinder | U.S. National Library of Medicine | RRID:SCR_016643 | For ORF identification |
| Software, algorithm | NCBI CD-Search | U.S. National Library of Medicine | | For mucin domain search |
| Software, algorithm | Pfam 31.0 | DOI: 10.1093/nar/gkv1344 | RRID:SCR_004726 | For mucin domain search |
| Software, algorithm | SMART | DOI: 10.1093/nar/gkx922 | RRID:SCR_005026 | For mucin domain search |

*Continued on next page*

*Continued*

| Reagent type (species) or resource | Designation | Source or reference | Identifiers | Additional information |
|---|---|---|---|---|
| Software, algorithm | Zen (version 11.0.3. 190 2012-SP2) | Zeiss | RRID: SCR_013672 | For microscope images |
| Software, algorithm | ImageJ (1.51 k) | DOI: 10.1038/nmeth.2089 | RRID: SCR_002285 | For area and length analysis |
| Software, algorithm | R Studio (1.2.1335) | RStudio, Inc | RRID: SCR_000432 | For bioinformatics |
| Software, algorithm | Prism (v8.3.0) | GraphPad | RRID: SCR_002798 | Graphing |

## Planarian maintenance and care

Asexual *Schmidtea mediterranea* (clonal line CIW4, RRID:NCBITaxon:79327) (*Sánchez Alvarado et al., 2002*) were maintained in 0.5 g/L Instant Ocean salts with 0.0167 g/L sodium bicarbonate dissolved in Type I water (*Roberts-Galbraith and Newmark, 2013*), and fed with beef liver paste. For all experiments, planarians were starved seven days prior to fixation. For LCM, planarians were 6–9 mm in length. For WISH and FISH, planarians were 2–4 mm in length. All animals were randomly selected from large (300–500 animals) pools, with the exception that animals with blastemas (e.g. those that had recently fissioned) were excluded.

## Optimization of planarian fixation for RNA extraction and histology

Planarians were relaxed in 0.66 M $MgCl_2$ or treated with 7.5% N-acetyl-L-cysteine or 2% HCl (ice-cold) for 1 min to remove mucus as described (*Forsthoefel et al., 2014*). Planarians were fixed in 4% formaldehyde/1X PBS, or methacarn (6 mL methanol, 3 mL chloroform, 1 mL glacial acetic acid) for 10 min at room temperature as described (*Forsthoefel et al., 2014*). Formaldehyde-fixed samples were washed three times (5 min each) in 1X PBS. Methacarn-fixed samples were first rinsed three times in methanol, then rehydrated in 1:1 methanol:PBS for 5 min, followed by three washes (5 min each) in PBS. For analysis of RNA integrity, 5–10 planarians were immediately homogenized in Trizol, and RNA was extracted using two chloroform extractions and high-salt precipitation buffer according to the manufacturer's instructions. RNA samples were analyzed using Agilent RNA Screen-Tape on an Agilent TapeStation 2200 according to the manufacturer's protocol.

For histology on methacarn-fixed samples, animals were relaxed and fixed individually in glass vials to minimize adherence to other samples. After fixation and rehydration, animals were incubated in 5%, 15%, and 30% sucrose (in RNAse-free 1X PBS), for 5–10 min each. Samples were then mounted and frozen in OCT medium, and cryosectioned at 20 µm thickness onto either Superfrost Plus glass slides (Fisher 12-550-15) (for staining optimization) or PEN membrane slides (Fisher/Leica No. 11505158) (for LCM). Prior to cryosectioning, PEN membrane slides were treated for 1 min with RNAse*ZAP* (Invitrogen AM9780), then rinsed by dipping 10 times (1–2 s per dip) in three successive conical tubes filled with 30 mL DEPC-treated water, followed by 10 dips in 95% ethanol and air-drying for 5–10 min. After cryosectioning, slides were stored on dry ice for 2–4 hr prior to staining. Cryostat stage and blades were wiped with 100% ethanol prior to sectioning.

For histological staining, slides were warmed to room temp. for 5–10 min. All slides were stained individually in RNAse-free conical tubes by manually dipping (1–2 s per dip) in 30 mL solutions. Forceps used for dipping were treated with RNAse*ZAP* and ethanol. All ethanol solutions were made with 200 proof ethanol and DEPC-treated water.

For Hematoxylin staining: 70% ethanol (20 dips); DEPC-treated water (20 dips); Mayer's Hematoxylin (Sigma Aldrich MHS16-500ML) (15 dips); DEPC-treated water (10 dips); Scott's Tap Water (2 g sodium bicarbonate plus 10 g anhydrous $MgSO_4$ per liter of nuclease-free water) (10 dips); 70% ethanol (10 dips); 95% ethanol (10 dips); 95% ethanol (10 dips); 100% ethanol.

For Eosin Y staining: 70% ethanol (20 dips); DEPC-treated water (20 dips); 70% ethanol (20 dips); Alcoholic Eosin Y (Sigma Aldrich HT110116-500ML) (100%, 10%, or 2% diluted into 200 proof ethanol) (15 dips); 95% ethanol (10 dips); 95% ethanol (10 dips); 100% ethanol. In some cases, fewer dips (8-10) in Eosin Y were required for better differentiation of gut tissue.

For combined Hematoxylin and Eosin Y staining: 70% ethanol (20 dips); DEPC-treated water (20 dips); Mayer's Hematoxylin (15 dips); DEPC-treated water (10 dips); Scott's Tap Water (10 dips); 70% ethanol (10 dips); 10% Alcoholic Eosin Y (15 dips); 95% ethanol (10 dips); 95% ethanol (10 dips); 100% ethanol.

The entire staining protocol was completed in less than 5 min. Slides were air dried for 5 min, then stored in plastic slide boxes on dry ice for 2–4 hr before LCM. Although we tested overnight storage at −80°C, we found that section morphology and RNA quality were best when conducting all steps, from fixation to LCM, on the same day.

## Laser-capture microdissection and RNA extraction

Stained PEN slides were removed from dry ice and immediately immersed for 30 s in ice-cold 100% ethanol, then room temperature 100% ethanol to minimize condensation/rehydration of sections and maintain RNAse inactivation during warming. Slides were then air dried for 2–3 min, and mounted in a Leica LMD7 laser microdissection microscope. Samples were dissected at 10X magnification using the following parameters: Power-30; Aperture-20; Speed-5; Specimen Balance-1; Head Current-100%; Pulse Frequency-392 Hz. Regions were dissected into empty RNAse-free 0.5 mL microcentrifuge caps (Axygen PCR-05-C). We separately captured medial intestine, lateral intestine, and non-intestine regions from all sections (8-10) per slide within 45–50 min. After capture, 20 µL Buffer XB (Arcturus PicoPure RNA Isolation Kit 12204–1) was added to captured tissue, then tubes were immediately frozen on dry ice and stored at −80°C prior to RNA extraction.

For RNA extraction, samples were thawed for 5 min at room temperature. Next, tissue from two tubes/slides (16–20 sections from the same planarian) was pooled for each biological replicate, incubated at 42°C for 30 min, and RNA was extracted using the Arcturus PicoPure RNA Isolation Kit following the manufacturer's instructions. 40–400 ng of total RNA was obtained from each sample, measured using a Denovix UV Spectrophotometer. RNA quality was analyzed using Agilent HS RNA ScreenTape on an Agilent TapeStation 2200 according to the manufacturer's protocol.

## Library preparation and RNA sequencing

Concentration of RNA was ascertained using a Thermo Fisher Qubit fluorometer. RNA quality was verified using the Agilent Tapestation. Libraries were generated using the Lexogen Quantseq 3' mRNA Library Prep Kit according to the manufacturer's protocol, with 5 ng total RNA input for each sample. Briefly, first-strand cDNA was generated using 5'-tagged poly-T oligomer primers. Following RNase digestion, second strand cDNA was generated using 5'-tagged random primers. A subsequent PCR step with additional primers added the complete adapter sequence to the initial 5' tags, added unique indices for demultiplexing of samples, and amplified the library. Final libraries for each sample were assayed on the Agilent Tapestation for appropriate size and quantity. Libraries were then pooled in equimolar amounts as ascertained by fluorometric analysis. Final pools were quantified using qPCR on a Roche LightCycler 480 instrument with Kapa Biosystems Illumina Library Quantification reagents. Sequencing was performed using custom primers on an Illumina Nextseq 500 instrument with High Output chemistry and 75 bp single-ended reads.

## Short-read mapping and gene-expression analysis

Adapters and low quality reads were trimmed from fastq sequence files with BBDuk (https://source-forge.net/projects/bbmap/, RRID:SCR_016969) using Lexogen data analysis recommendations (https://www.lexogen.com/quantseq-data-analysis/): k = 13 ktrim = r forcetrimleft = 11 useshortkmers = t mink = 5 qtrim = t trimq = 10 minlength = 20. Sequence quality was assessed before and after trimming using FastQC (RRID:SCR_014583) (*Andrews, 2010*). Reads were then mapped to a version of the de novo dd_Smed_v6 transcriptome (*Brandl et al., 2016*) restricted to 28,069 unique transcripts (i.e., those transcripts whose identifiers ended with the suffix '_1') using Bowtie2 (v2.3.1) (*Langmead and Salzberg, 2012*) with default settings. Resulting SAM files were converted to BAM files, sorted, and indexed using Samtools (v1.3, RRID:SCR_002105) (*Li et al., 2009*). Raw read counts per transcript were then generated for each BAM file using the 'idxtats' command in Samtools, and consolidated into a single Excel spreadsheet.

The resulting read counts matrix was imported into R (RRID:SCR_000432), then analyzed in edgeR v3.8.6 (RRID:SCR_012802) (*Robinson et al., 2010*). First, all transcripts with counts per million

(CPM) <1 in 4/12 samples (e.g. lowly expressed transcripts) were excluded from further analysis (13,136/28,069 transcripts were retained). Next, after recalculation of library size, samples were normalized using trimmed mean of M-values (TMM) method, followed by calculation of common, trended, and tagwise dispersions. Finally, differentially expressed transcripts in intestinal vs. non-intestinal samples were determined using the generalized linear model (GLM) likelihood ratio test. 1911 transcripts had a fold-change of more than 2 (logFC >1) and an FDR-adjusted p value < 0.01 in either medial or lateral intestine, relative to non-intestinal tissue. We further limited to 1844 transcripts with a minimum transcripts-per-million (TPM) of 2 in 4 of any eight intestinal biological replicates (medial or lateral), since transcripts with lower expression values were at the lower limit of detection by ISH, and their removal also modestly increased the robustness of LCM vs. phagocyte analysis.

## Human protein atlas comparison

We queried (TBLASTX, RRID:SCR_011823) 28,069 unique dd_Smed_v6 nucleotide sequences against 20,726 nucleotide sequences in the human reference proteome downloaded from UniProt (www.uni-prot.org) (Release 2017_12, 20-Dec-2017). 13,362 dd_Smed_v6 transcripts hit human sequences (7309 unique) with $E$-value $\leq 1 \times 10^{-3}$. These human sequences were then used to conduct reciprocal TBLASTX queries against dd_Smed_v6 transcripts: 7220 hit 5808 unique dd_Smed_v6 sequences with $E$-value $\leq 1 \times 10^{-3}$. In total, 5583 dd_Smed_v6 transcripts had reciprocal best hits (RBHs) in the human proteome, with >94% having $E$-value $\leq 1 \times 10^{-10}$ in either direction. Of 1844 intestine-enriched transcripts, 701 had RBHs in the human proteome.

Next, using RBH UniProt Accession numbers, we extracted RNA-Seq tissue enrichment data from the Human Protein Atlas (*Uhlén et al., 2015*). 699/701 intestinal transcripts' RBH homologs were present in the HPA data; of these, 130 were enriched in one or more human tissues. 5,561/5,583 dd_Smed_v6 transcripts' RBH homologs were present in the HPA data; of these, 1011 were enriched in one or more human tissues. The number of intestine and non-intestine RBH homologs enriched in each of 32 human tissues was calculated, and expressed as a percentage of total transcripts (699 intestine or 5561 non-intestine) with RBH homologs. Finally, ratio of the percentage of intestine-enriched RBH homologs to the percentage of all dd_Smed_v6 RBH homologs was then calculated to determine fold-enrichment for each tissue. Two tissues ('Appendix' and 'Smooth Muscle') were excluded from final analysis since <0.1% (6/5561) of all dd_Smed_v6 transcripts had RBH homologs enriched in these tissues.

TBLASTX queries were conducted using NCBI BLAST+ standalone suite. Extraction and analysis of tissue enrichment data from HPA was conducted in R and Excel.

## Gene Ontology annotation, nomenclature, and analysis

BLASTX (RRID:SCR_001653) homology searches of UniProtKB protein sequences for *H. sapiens, M. musculus, D. rerio, D. melanogaster,* and *C. elegans* were conducted using all 28,069 unique transcripts in the 'dd_Smed_v6' transcriptome in PlanMine as queries. In all figures, gene names/abbreviations are based on the best (lowest $E$-value) UniProt homolog, except: (1) genes were named after the best human homolog for HPA analysis; and (2) we used *Smed* nomenclature when genes (or paralogs) were previously named by us or others, including *hnf-4*/dd_1694_0_1 (43), *nkx2.2*/dd_2716_0_1 (34), *gata4/5/6*/dd_4075_0_1 (43), *apob-1*/dd_636_0_1 [DJF, unpublished], *apob-2*/dd_194_0_1 [DJF, unpublished], *slc22a6*/dd_1159_0_1 (75), *gli-1*/dd_7470_0_1 (82), and *RREB2*/dd_10103_0_1 (134). Biological Process GO terms (also obtained from UniProtKB) from the top hit for each species (with $E$-value $\leq 1 \times 10^{-5}$) were assigned to each dd_Smed_v6 transcript. 9344 of 28,069 total transcripts and 1379 of 1844 intestine-enriched transcripts were annotated with GO terms. Enrichment for specific terms among all intestine-enriched transcripts, medially enriched transcripts, or laterally enriched transcripts was then evaluated with BiNGO (RRID:SCR_005736) (*Maere et al., 2005*). Regional transcript enrichment was calculated as a ratio of fold changes in medial and lateral intestinal tissue: transcripts were considered to be medially enriched if $FC_{medial}/FC_{lateral} > 1$ (1221 transcripts), or laterally enriched if $FC_{medial}/FC_{lateral} < 1$ (623 transcripts). All 13,136 (of 28,069) transcripts detected in our experiment (above) were used as a background set, and hypergeometric testing with a Benjamini and Hochberg False Discovery Rate (FDR) of 0.05 was considered significant. We additionally restricted our summarization to terms that were annotated to

more than one percent of transcripts that received annotations in each group: 1379/1844 intestine-enriched transcripts, 924/1221 medially enriched transcripts, or 455/623 laterally enriched transcripts.

### Comparison to gene sets involved in innate immunity

1456 *Dugesia japonica* transcripts upregulated in response to either *L. pneumophila* or *S. aureus* infection (*Abnave et al., 2014*) were queried (TBLASTX) against 28,069 dd_Smed_v6_unique transcripts. 981 dd_smed_v6 transcripts hit with evalues < 1e-03. After removing duplicate hits, 783 transcripts remained. 701/783 transcripts were present in the full 13,136 LCM dataset; 99/783 were intestine-enriched.

30,021 SMED_20140614 transcripts were mapped to 28,069 dd_smed_v6_unique transcripts, generating 22,889 dd_smed_v6_unique hits with evalues < 1e-03. After removing duplicates, 19,338 transcripts remained. 741/19,338 transcripts were up- or down-regulated in planarians shifted to static culture (*Arnold et al., 2016*). 447/741 transcripts were present in the full 13,136 LCM dataset, and 62 were intestine-enriched.

### Comparison to gene expression in sorted phagocytes

28,069 unique dd_Smed_v6 transcripts were blasted (BLASTN) against 11,589 ESTs and assembled contigs from the 'SmedESTs3' collection (*Zayas et al., 2005*), which were used in microarray-based analysis of gene expression in sorted phagocytes (*Forsthoefel et al., 2012*). 7927 hit with length >100 bp, >90% base identity, and *E*-value less than $1 \times 10^{-50}$. Of these, 6919 of 13,136 Smed_v6 transcripts detected in LCM samples mapped to unique Smed_ESTs3 contigs or ESTs with detectable expression in the sorted phagocyte data set. 2626/6919 transcripts had FDR-adjusted p values < 0.01 for logFC in either medial or lateral intestine (vs. non-intestine, LCM data in this study), and 1498/2626 had FDR-adjusted p values < 0.05 for logFC in sorted phagocytes (vs. all other cells) (phagocyte data in *Forsthoefel et al., 2012*). 1317/2626 had logFC >1 in either medial or lateral intestine. In the sorted phagocyte data, 900/1317 had a fold-change >0 and FDR-adjusted p value < 0.05 ('high' in phagocytes), 358/1317 had an FDR-adjusted p value > 0.05 ('moderate' in phagocytes), and 59/1317 had a fold-change <0 and FDR-adjusted p value < 0.05 ('low' in phagocytes). For *Figure 5B*, we identified 1067/1514 phagocyte-enriched transcripts with corresponding dd_Smed_v6 transcripts, including some with logFC 0.6–1, as in the original study (*Forsthoefel et al., 2012*).

### Comparison to gene expression in single-cell transcriptomes

We utilized data from gene expression analysis of single planarian cells from two recent studies (*Fincher et al., 2018*; *Plass et al., 2018*) to identify dd_Smed_v6 transcripts enriched in specific intestinal cell types that were also represented in our 1844 intestine-enriched transcripts (this study) and phagocyte expression data (*Forsthoefel et al., 2012*; *Figure 2—figure supplement 1*). For comparison to *Fincher et al. (2018)*, we identified 391 phagocyte-enriched transcripts (subcluster four or 'enterocytes' in Table S2 (intestine); *Fincher et al., 2018*), 21 goblet-enriched transcripts (subcluster eight in *Fincher et al., 2018*), and 30 basal-enriched transcripts (subcluster eight or 'outer intestinal cells' in *Fincher et al., 2018*); transcripts found in other intestinal subclusters were excluded. For comparison to *Plass et al. (2018)*, we identified 92 phagocyte-enriched (but not goblet-enriched) transcripts and 27 goblet cell-enriched (but not phagocyte-enriched) transcripts that were also represented in our 1844 intestine-enriched transcripts and phagocyte expression data. For global comparison of transcripts enriched in laser captured intestine (*Figure 5*), we included all transcripts enriched in intestinal cell types (and intestinal precursors) in single cell studies (*Fincher et al., 2018*; *Plass et al., 2018*; *Swapna et al., 2018*), without regard to cell type or enrichment in non-intestinal cell types. For comparison to *Swapna et al. (2018)* data, BLASTN was used to identify dd_Smed_v6 transcripts orthologous to 'SmedASXL' transcripts (with >95% identity and length >100 bp).

### Gene cloning and expressed sequence tags

Total RNA was extracted from planarians using Trizol with two chloroform extractions and high salt precipitation. After DNAse digestion, cDNA was synthesized using the iScript Kit (Bio-Rad 1708891).

Genes were amplified by PCR with Platinum *Taq* (Invitrogen 10966026) using primers listed in *Supplementary file 1*. Amplicons were cloned into pJC53.2 (RRID:Addgene_26536) digested with Eam1105I as described (*Collins et al., 2010*) and sequenced to verify clone identity and orientation. For some genes, expressed sequence tags (ESTs) in pBluescript II SK(+) were utilized (*Zayas et al., 2005*), also listed in *Supplementary file 1*.

## In situ hybridization and immunofluorescence

In situ hybridizations were performed as described (*King and Newmark, 2013*), with the following adjustments: NAc (7.5%) treatment was for 15 min; 4% formaldehyde fixation was for 15 min; and animals were bleached for 3 hr. Samples were pre-incubated with tyramide solution without $H_2O_2$ for 10 min, then spiked with $H_2O_2$ (.0003% final concentration), then developed for 10 min. Immunofluorescence with mAb 6G10 after FISH was conducted as described (*Ross et al., 2015*), using 1x PBS, 0.3% Trition-X100, 0.6% IgG-free BSA, 0.45% fish gelatin as the blocking buffer (*Forsthoefel et al., 2014*). Images are representative of two independent in situ hybridizations on 4–6 animals per experiment.

## Mucin identification

*S. mediterranea* mucin-like genes in PlanMine (*Brandl et al., 2016*) and SmedGD2.0 (20) were identified by TBLASTN searches with human refseq_protein and UniProt mucin sequences. Planarian sequences were translated with NCBI ORFinder (https://www.ncbi.nlm.nih.gov/orffinder/, RRID:SCR_016643), and domain searches were conducted using NCBI CD-Search (https://www.ncbi.nlm.nih.gov/Structure/cdd/cdd.shtml) (*Marchler-Bauer et al., 2015*), Pfam 31.0 (https://pfam.xfam.org, RRID:SCR_004726) (*Finn et al., 2016*), and SMART (http://smart.embl-heidelberg.de, RRID:SCR_005026) (*Letunic and Bork, 2018*). Three planarian sequences were identified that encoded three N-terminal von Willebrand factor D domains and two or more N-terminal cysteine-rich and trypsin-inhibitor-like cysteine-rich domains characteristic of human mucins (e.g. MUC-2, MUC-5AC) (*Lang et al., 2007*; *Lang et al., 2016*), but not von Willebrand factor A or Thrombospondin type I repeats found in closely related proteins (e.g. SCO-spondin). Planarian mucin-like genes identified using this approach are: *Smed-muc-like-1* (dd_Smed_v6_17988_0_1/SMED30009111), *Smed-muc-like-2* (dd_Smed_v6_18786_0_1/SMED30002668), and *Smed-muc-like-3* (dd_Smed_v6_21309_0_1/dd_Smed_v6_38233_0_1/dd_Smed_35076_0_1/SMED30006765).

## Identification of intestine-enriched transcription factors and RNA interference

Putative transcription factors were identified by extracting intestine-enriched transcripts with 'DNA' and/or 'transcription' Biological Process and Molecular Function GO terms, then verifying that the best Uniprot homologs regulated transcription in published experimental evidence. *Smed-gli-1* has been previously studied (*Rink et al., 2009*). *Smed-RREB2* was most homologous to another planarian zinc finger protein, *Smed-RREBP1* (134). However, we used the more common '*RREB*' abbreviation (rather than '*RREBP*') for the second planarian paralog.

RNAi experiments were conducted as described (*Rouhana et al., 2013*) by mixing one microgram of in vitro-synthesized dsRNA with 1 µL of food coloring, 8 µL of water, and 40 µL of 2:1 liver:water homogenate. For the primary regeneration screen, 10 animals were fed three times over 6 days, amputated 4–5 days after the last feeding, then fixed 6 days post amputation. For further analysis of *gli-1(RNAi)* and *RREB2(RNAi)* phenotypes in uninjured animals, planarians were fed 1X/week for 6 weeks or 2X/week for 3 weeks (six feedings total). *gli-1(RNAi)* and *RREB2(RNAi)* animals were scored as 'non-eaters' if they refused food on two successive days (7 and 8 days after the previous feeding), and were removed the experiment. Non-eating and/or curling animals were excluded from FISH analysis. For experiments in regenerates, animals were fed 2X/week for 4 weeks (eight feedings total). *egfp* dsRNA (*Forsthoefel et al., 2012*) was used as the negative control in all experiments.

## Image collection

Confocal images were collected on a Zeiss 710 Confocal microscope, with the following settings: two tracks were used, one with both 405/445 (excitation/emission nm, blue) and 565/650 (red), and the other with only 501/540 (green) to minimize bleed-through between channels. Detector gains

were adjusted so that no pixels were saturated. Digital offset was set to 0 or −1, to ensure that most pixel intensities were non-zero, with averaging set to 2. Whole animals were captured with a single z-plane (5 μm section) with a 10x objective, tiled, and stitched in Zen (version 11.0.3.190, 2012-SP2, RRID:SCR_013672). Magnified regions were captured with a 63x Oil immersion objective and a z-stack (50–100 slices, 0.31 μm/slice) was collected from the tail and/or head of the animal. In some cases, after imaging min/max or linear best fit adjustments were made in Zen to improve contrast of final images. Profile graphs represent raw, unadjusted pixel intensity values from a single optical section.

WISH images were collected on a Zeiss Stemi 508 with an Axiocam 105 color camera, an Olympus SZX12 dissection microscope with an Axiocam MRc color camera, or a Zeiss Axio Zoom.V16 with an Axiocam 105 camera. In some cases, brightness and/or contrast were adjusted in photoshop to improve signal contrast.

## Quantification of animal area and length for *gli-1* and *RREB2* knockdown animals

Animals were separated into 35 mm x 10 mm petri dishes in groups of two, and imaged on a Zeiss Stemi 508 prior to the first dsRNA feeding, then again seven days after the fifth feeding (animals were fed once per week). For area, images were processed with ImageJ (RRID:SCR_002285) (*Schneider et al., 2012*) by first applying an Auto Threshold, using method = Intermodes, Huang, or Triangles with 'White objects on black background' selected, to highlight the planarian. Analyze Particles was then run, with size = 600000–10000000 $\mu m^2$, and 'Display results' and 'Include holes' selected. For length, a straight line was drawn manually from the tip of the head to the tip of the tail and then Measure was used to obtain the length. Numerical data were analyzed and plots generated in GraphPad Prism v8.3.0 (RRID:SCR_002798).

## Ethics statement

No vertebrate organisms were used in this study.

# Acknowledgements

We thank Forrest Waters for initial optimization of fixation, labeling, and RNA extraction for laser microdissection. We thank Mayandi Sivagaru (Institute for Genomic Biology, University of Illinois at Urbana-Champaign) and Muralidharan Jayaraman (University of Oklahoma Health Sciences Center) for help with LCM optimization, and Alvaro Hernandez (Roy J Carver Biotechnology Center, Illinois) and Graham Wiley (OMRF Clinical Genomics Center) for library preparation and Illumina sequencing. We thank Ben Fowler, Amanda Valdez, Julie Crane, and Alex Weddle (OMRF Imaging Core) for technical assistance with confocal microscopy, histology, and LCM slide preparation; Jonathan Wren (OMRF) (Bioinformatics and Pathways Core) for advice on reciprocal blast searches; and Stuart Glenn and the OMRF Quantitative Analysis Core for high-performance computing resources. We thank Jenny Lee (OMRF) for illustrations of intestinal cell types in *Figure 9*, and Josh Sisson (OMRF) for the PlanGut website design. We are grateful to Ricardo Zayas and Francesc Cebrià, who originally identified *Smed-npc2* as a marker for goblet cells. We gratefully acknowledge members of the Newmark and Forsthoefel Labs, Linda Thompson, Dean Dawson, Gaurav Varshney, Wan Hee Yoon, Jian Li, and David Jones for discussions and manuscript critique. The OMRF Imaging Core and Bioinformatics and Pathways Core were supported by NIH/COBRE GM103636. The OMRF Quantitative Analysis Core was supported by NIH/COBRE GM110766. LMD was supported by the Molecular Biology Shared Resource, Stephenson Cancer Center (SCC) CCSG P30CA225520 and SCC COBRE P20GM103639. PAN is an investigator of the Howard Hughes Medical Institute.

# Additional information

### Competing interests

Phillip A Newmark: Reviewing Editor, eLife. The other authors declare that no competing interests exist.

## Funding

| Funder | Grant reference number | Author |
| --- | --- | --- |
| Oklahoma Center for Adult Stem Cell Research | 4340 | David J Forsthoefel |
| National Institute of General Medical Sciences | COBRE GM103636-Project 1 | David J Forsthoefel |
| National Institute of Child Health and Human Development | HD043403 | Phillip A Newmark |
| Howard Hughes Medical Institute | | Phillip A Newmark |

The funders had no role in study design, data collection and interpretation, or the decision to submit the work for publication.

## Author contributions

David J Forsthoefel, Conceptualization, Formal analysis, Supervision, Funding acquisition, Validation, Investigation, Visualization, Methodology, Project administration; Nicholas I Cejda, Conceptualization, Data curation, Formal analysis, Validation, Investigation, Visualization, Methodology; Umair W Khan, Conceptualization, Methodology; Phillip A Newmark, Conceptualization, Supervision, Funding acquisition, Project administration

## Author ORCIDs

David J Forsthoefel (ID) https://orcid.org/0000-0002-8583-4383
Nicholas I Cejda (ID) https://orcid.org/0000-0003-4518-4125
Umair W Khan (ID) http://orcid.org/0000-0003-0206-0667
Phillip A Newmark (ID) https://orcid.org/0000-0003-0793-022X

## Decision letter and Author response

Decision letter https://doi.org/10.7554/eLife.52613.sa1
Author response https://doi.org/10.7554/eLife.52613.sa2

# Additional files

## Supplementary files

• Supplementary file 1. RNA-Seq and other data for 1,844 intestine-enriched transcripts. (S1A) Summary of RNA-Seq data and whole-animal in situ expression patterns for 1844 transcripts enriched in laser-captured planarian intestine, together with phagocyte expression, single-cell RNA-Seq data, best BLAST hits for human/mouse/zebrafish/fly/*C. elegans*, cloning primers, and transcript sequences. Single-cell data from *Plass et al. (2018)* reprinted with permission from AAAS. (S1B) Figure locations for transcript expression patterns shown in the manuscript.

• Supplementary file 2. RNA-Seq and other data for all 13,136 transcripts detected in laser-captured tissue. Summary of RNA-Seq data, together with phagocyte expression, single-cell RNA-Seq data, best BLAST hits for human/mouse/zebrafish/fly/*C. elegans,* and transcript sequences. Single-cell data from *Plass et al. (2018)* reprinted with permission from AAAS.

• Supplementary file 3. Intestinal transcripts in Gene Ontology categories. (S3A) Examples of intestine-enriched transcripts (LCM) annotated with digestive physiology-related GO Biological Process Terms. (S3B) Transcripts predicted to encode intestine-enriched solute carrier proteins (SLCs). (S3C) Transcripts predicted to encode intestine-enriched Rab GTPases. (S3D) Transcripts predicted to encode intestine-enriched tumor necrosis factor receptor associated factors (TRAFs). Single cell data from *Plass et al. (2018)* reprinted with permission from AAAS.

• Supplementary file 4. Intestinal transcripts with human homologs enriched in digestive tissues. Examples of intestine-enriched transcripts (LCM) with human RBH homologs enriched in human digestive tissues at the transcript level (*Uhlén et al., 2015*). Transcript entries include LCM RNA-Seq

data, together with WISH patterns, phagocyte expression, and single-cell RNA-Seq data. Single-cell data from *Plass et al. (2018)* reprinted with permission from AAAS.

• Supplementary file 5. RNA interference results for intestine-enriched transcription factors and mediolaterally enriched transcripts. (S5A) RNAi results for medially enriched transcripts. (S5B) RNAi results for laterally enriched transcripts. (S5C) RNAi results for transcription factors.

• Transparent reporting form

## Data availability

Raw and processed RNA-Seq data associated with this study have been deposited in the NCBI Gene Expression Omnibus (GEO) under accession number GSE135351.

The following dataset was generated:

| Author(s) | Year | Dataset title | Dataset URL | Database and Identifier |
|---|---|---|---|---|
| Forsthoefel DJ, Cejda NI, Khan UW, Newmark PA | 2019 | Transcriptional profiling of laser captured intestinal tissue from the planarian Schmidtea mediterranea | https://www.ncbi.nlm. nih.gov/geo/query/acc. cgi?acc=GSE135351 | NCBI Gene Expression Omnibus, GSE135351 |

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
