## [Decision Letter]

**Acceptance summary:**

Using laser capture microdissection followed by RNA sequencing, the authors identify novel transcripts expressed in specific locations within the planarian gut, revealing an unappreciated spatial complexity, something that cannot be easily captured by the much-used single cell RNA sequencing approach. This is a useful method for the community, and serves as an important reminder that caution must be exercised when relying only on scRNA-seq.

**Decision letter after peer review:**

Thank you for submitting your article "Cell-type diversity and regionalized gene expression in the planarian intestine" for consideration by *eLife*. Your article has been reviewed by three peer reviewers, and the evaluation has been overseen by a Reviewing Editor and Naama Barkai as the Senior Editor. The reviewers have opted to remain anonymous.

The reviewers have discussed the reviews with one another and the Reviewing Editor has drafted this decision to help you prepare a revised submission.

Summary:

Using laser capture microdissection followed by RNA sequencing, the authors determine novel transcripts expressed in specific locations within the planarian gut, revealing an unappreciated spatial complexity, something that cannot be captured by much-used single cell RNA sequencing. This is a useful method for the community, and reveals a critical caveat in relying on scRNA seq.

Essential revisions:

In the course of discussion, it became clear that, whereas the method part is very solid, the gene list part can be presented in a more useful way, for example excel list that integrates both LCM-RNA-seq and existing scRNA-seq results (or even better is a website, but the reviewers agreed that it might be too much to ask). Without making the data set useful, the potential value of this comprehensive work may not be realized to its full potential. During discussions among reviewers, they noted a few possible ways to make the data set more useful, which is provided below for authors' consideration.

"Perhaps the resource would be more useful if they could combine into one database (eg, an excel list, or website, etc) the catalog of intestine genes with predicted RNAseq enrichment and also and verified by WISH/FISH. It seems from the text that there are a large number of transcripts observed to be intestine enriched from the scRNAseq Fincher dataset that they do not recover in their method. I imagine that many of these do not have expression exclusive to the intestine, which to me highlights the utility of the current study. I don't believe the existing scRNAseq resources make the distinction between enriched versus exclusive expression, which as a user of them I have found to be limiting or at least a consideration in their utility. What is the real false-discovery rate in these enrichment lists? There would also be an opportunity here to clarify in one resource which transcripts have verified expression in the intestine (from this study and others) that could comment on this. My sense is that this effort could be completed within 2 months."

"I think integrating the new data with existing single-cell datasets indeed would be very useful. As the percentage of confirmed intestinally enriched transcripts from this laser-capture dataset is very high and the RNA concentrations are solid, I trust that this data will be more reproducible than that of the published sc datasets. Similar to others I have found that the raw gene lists from those studies are of limited use because genes listed in a tissue are not necessarily unique or even really enriched. Comparison between the various gene lists may give a better insight in how to filter or interpret single cell data versus actual tissue isolation data."

"The authors really need to focus on putting their data into the context of the scRNAseq studies that have been performed. "

Individual reviews are show below for the authors to address minor points (editing for clarity).

Reviewer #1:

In the manuscript by Forsthoefel et al., the authors use laser capture microdissection with RNA sequencing to determine novel transcripts expressed in specific spatial locations in the planarian gut. The authors find an unappreciated spatial complexity of planarian gut gene expression, but also perform some comparative gene expression from the mammalian endodermal structures. The authors then dissect the function of a *gli* and *RREB2* transcription factors in regulating goblet cell numbers. The manuscript is well-written and the experiments are of high quality. Below are a couple of concerns the authors might want to consider.

1) On one hand, this manuscript is aiming to be a resource paper, and thus, it reads a bit like a laundry list of genes found. It is hard to put the current study in the context of previous ones, particularly the single-cell atlas-type papers. It would have been nice to contrast and compare what the authors found that the other studies missed, or why their method is better, etc.

2) While the experiments and analyses are well done, it is hard to imagine the current study being widely used by planarian researchers or in the gut community as a whole. Part of the problem is that the data are not presented in a way that is easy for others to interact with them as a resource to come back to.

Reviewer #2:

This report by Forsthoefel, Newmark and colleagues implements a novel laser capture microdissection and RNAseq method to significantly clarify the cell types and gene expression within the planarian intestine and identify new intestinal regulators that participate in regeneration. Though at large, much emphasis has been placed on identifying cell-specific transcriptomes through scRNAseq, defining the spatial nature of cell transcriptomes is a critical ongoing challenge, and this report rigorously shows the utility of LCM for this purpose in planarians. The authors first optimize the LCM process by performing a rigorous and extensive comparison of the influence of fixation and H&E staining on RNA extraction from tissue sections, then use this assay combined with RNAseq along with WISH/FISH validation to identify a comprehensive set of genes expressed in different regions of the intestine. Surprisingly, this effort identified many factors that clarify a subdivision of intestinal cell states along the media-lateral axis and also point to specialization within the anterior and posterior intestine branches. Very intriguingly, a comparison of planarian intestine-specific transcripts to organ enrichment of human transcripts found an enrichment for digestive and energy storage/metabolism. A comparison with the author's prior work to transcriptionally profile enriched phagocyte cells led to identification of intestinal transcripts labeling other cell types, including goblet cells and basal cells. Finally, the authors conduct an RNAi screen of intestine-enriched transcription factors to identify genes important for goblet cell formation and function. They find a role for *gli-1* in forming newly regenerated secretory goblet cells in blastemas, and by contrast, find a role for the *RREB2* gene in maintaining goblet cells away from injury sites or homeostatically. At long times of inhibition, *gli-1* and *RREB2* RNAi animals have defects in feeding, size and viability. Together this is a very nicely written and thorough analysis, making a major step forward in understanding both the anatomy, gene expression, and gene function within the regenerative planarian intestine.

1) I would suggest some clarification in the text about the description of the *gli-1* and *RREB2* RNAi phenotypes. In analysis of *gli1*(RNAi), the text in paragraph two of subsection “Intestine-enriched transcription factors regulate goblet cell differentiation and maintenance” highlights the observation that this treatment causes depletion of goblet cells within blastemas. However, it is also noteworthy that these cells seem reduced within the pre-existing regions as well and are also reduced in uninjured animals within the head and tail. This reading then sets up a comparison with the *RREB2*(RNAi) phenotype in which goblet cells from pre-existing regions were strongly reduced. However, *RREB2*(RNAi) goblet cells do not seem to have a normal abundance in the head blastema (Figure 7A/B), despite the description in the text ("arose normally"). I agree with the suggestion that the two genes could be subfunctionalized for generating versus maintaining goblet cells, but to me the evidence at hand does not present as much of an "all or nothing" case and so I find the term "mirror image" more a description of a model rather than definitively supported by the analysis. Though the acknowledgement of "complex observations" is appreciated here, some additional rewording could better align the text with the findings in the images or bring out these uncertainties. Alternatively, perhaps quantification or presenting other images from the animals probed could more clearly show the extent of distinctions between the phenotypes described in the text. Certainly, more experimentation on these two phenotypes could further clarify the function of the *RREB2* and *gli1* genes (eg, specificity for the intestine, effects on progenitor cells, direct tests of function on newly-formed versus pre-existing goblet cells, etc). However, I think it is beyond the scope of this report to embark on an extensive analysis of these two phenotypes at this time, as the work already makes a major contribution through the dataset for intestine cell transcriptomes, the LCM method, and the proof-of-principle use of the dataset for finding factors involved in intestinal biology.

2) The finding of shared intestinal expression between planarians and humans is very interesting. An aspect of this experiment that is less completely supported from the presentation was the assignment of gene orthology across these diverse species. For example, how were gene families handled in which there may not be a 1-to-1 orthology? (for example, Smed-piwi-1 top blastx hit in human genome is to HsHIWI, which itself reciprocally blasts to Smed-piwi-3 in the planarian genome, so presumably piwi-1 would not be present in the reciprocal best hits yet likely have some conserved molecular function in the piRNA pathway). Given the uncertainties in this process, including the orthology dataset would probably be the best way to allow readers to evaluate the strength of the orthology argument put forward here.

Reviewer #3:

Intestinal regeneration is widely conserved among metazoans. In many species intestinal cells are frequently replaced over the course of their life, and many are able to recover from minor injuries in their intestinal tracts. However how newly generated cells integrate into the existing intestinal structures is not well understood. Planarians are an excellent model to study this process as they exhibit turnover of their intestinal cells as well as extensive intestinal regeneration upon wounding. This manuscript describes the use of laser capture to isolate intestinal tissue for mRNA analysis to identify differential gene expression, and validates the method by confirming the expression of a large number of identified transcript by in situ hybridisation. The manuscript also identifies unexpected regional specialisation of the planarian intestine.

Overall the manuscript is well written and the approach is very thorough. The authors tested a number of different fixation strategies in order to get good RNA, and were able to identify a long list of intestinal transcripts. The number of verified transcripts is impressive, and the percentage of confirmed intestinal transcripts is very high, demonstrating that the method as presented works very well. The stainings are well-executed and the imaging is of very high quality. I have no technical concerns with any of the experiments.

This is a valuable new method for planarian community, and the optimisations described for the purpose of RNA analysis are likely to also be applicable to other model organisms, thereby making laser capture for RNA analysis more accessible to other communities as well. In addition to this technical advance, the manuscript delivers a better understanding of the cellular subtypes in the planarian intestine. In order for this to become accessible to the wider community it will be essential to incorporate a data file with actual sequences of the transcripts, or at least the primer sequences used for cloning, as no real consensus transcriptome currently exists and therefore transcript code numbers or gene names will not remain interpretable.

The only concern I have in terms of impact is that whereas the optimised laser-capture method is widely applicable, the use for the intestinal gene expression dataset is largely limited to the planarian community.

The organization of the manuscript however should be improved. The text is long and reads as a long list of gene names grouped by expected function. Some of the functional processes even come by multiple times over the course of the descriptions of the various stainings. As there is no data verifying the actual involvement of the intestine in these processes or further analysis of mechanistic aspects, this information could be equally well presented in a table. The primary information seems to be in Figure 1 which shows the method and the analysis, and Figure 7 which contains some very interesting new biology. The other figures and the text describing them could and should be condensed.

When turned into a short format, I think this is an interesting manuscript that makes a good case for laser-capture as a highly effective method to identify tissue-specific gene expression, and demonstrates that new biology can be uncovered by using it thoughtfully – as the authors clearly did.

---

## [Author Response]

Essential revisions:In the course of discussion, it became clear that, whereas the method part is very solid, the gene list part can be presented in a more useful way, for example excel list that integrates both LCM-RNA-seq and existing scRNA-seq results (or even better is a website, but the reviewers agreed that it might be too much to ask). Without making the data set useful, the potential value of this comprehensive work may not be realized to its full potential. During discussions among reviewers, they noted a few possible ways to make the data set more useful, which is provided below for authors' consideration."Perhaps the resource would be more useful if they could combine into one database (eg, an excel list, or website, etc) the catalog of intestine genes with predicted RNAseq enrichment and also and verified by WISH/FISH. It seems from the text that there are a large number of transcripts observed to be intestine enriched from the scRNAseq Fincher dataset that they do not recover in their method. I imagine that many of these do not have expression exclusive to the intestine, which to me highlights the utility of the current study. I don't believe the existing scRNAseq resources make the distinction between enriched versus exclusive expression, which as a user of them I have found to be limiting or at least a consideration in their utility. What is the real false-discovery rate in these enrichment lists? There would also be an opportunity here to clarify in one resource which transcripts have verified expression in the intestine (from this study and others) that could comment on this. My sense is that this effort could be completed within 2 months.""I think integrating the new data with existing single-cell datasets indeed would be very useful. As the percentage of confirmed intestinally enriched transcripts from this laser-capture dataset is very high and the RNA concentrations are solid, I trust that this data will be more reproducible than that of the published sc datasets. Similar to others I have found that the raw gene lists from those studies are of limited use because genes listed in a tissue are not necessarily unique or even really enriched. Comparison between the various gene lists may give a better insight in how to filter or interpret single cell data versus actual tissue isolation data.""The authors really need to focus on putting their data into the context of the scRNAseq studies that have been performed. "

We sincerely thank all three reviewers for their positive comments, and for raising the issue of data accessibility and recommending greater emphasis and clearer comparison with scRNA-Seq studies. We believe that addressing these concerns has significantly improved the manuscript.

First, we have developed a website, https://plangut.omrf.org, that enables user-friendly exploration of our laser-capture data, visualization of in situ expression patterns, and comparison of our results with recent scRNA-Seq studies and our previous phagocyte expression data. This web site includes essentially all the data in Supplementary files 1 and 2 (with the exception of five-organism BLAST homology), in a sortable and searchable format. Furthermore, one tab on the web page (Expression Patterns) enables visitors to click on rows to view 144 whole-mount in situ hybridization (WISH) expression patterns and compare with expression data from our group and others. We have referenced the website in our Results at appropriate locations. We also hope to include our data in future iterations of PlanMine (http://planmine.mpi-cbg.de/planmine/begin.do), a central repository for planarian genomic and transcriptomic data, to enable integration with the numerous resources there.

Second, we have re-organized several supplemental spreadsheets into Supplementary Files and Source Data, and we have also included more complete legends for all Supplementary files and Source data files. Originally, we provided RNA-Seq expression data, phagocyte expression data, descriptions of WISH expression patterns, cloning primers, and single cell RNA-Seq (scRNA-Seq) data, as Figure 1—source data 1. However, scRNA-Seq and other data were difficult to find in columns far to the right of the original spreadsheet, and we did not cross-reference all 13,136 transcripts detected by LCM with scRNA-Seq data. In the revised manuscript, we now provide a more concise version of Supplementary file 1 (1844 intestine-enriched transcripts), and a more comprehensive Supplementary file 2 (all 13,136 transcripts detected in laser-captured tissue). We have removed count data (CPM, RPKM, and TPM) from Supplementary file 1 for usability, and we have included sequences for all transcripts in both files, at the suggestion of reviewer #3.

Third, to increase the emphasis on comparison with scRNA-Seq studies, we have moved the comparison of our results with scRNA-Seq data to its own section in the Results ("Laser capture substantially increases resolution of the global intestinal transcriptome"). We have also developed a main text figure (Figure 5) to accompany this section. We have expanded this section to include Venn diagrams for individual cell types (Figure 5A), a comparison with our previous phagocyte data set (Figure 5B), a Venn diagram that globally compares scRNA-Seq with our data that was originally in a supplemental figure (now Figure 5C), and WISH images for more intestine-enriched transcripts that were uniquely detected in our study (Figure 5D-H). We have also added a sentence in this section of the Results speculating that fold-enrichment cutoffs might contribute to differences between data sets:

"The incomplete overlap between various scRNA-Seq studies and our results could be explained, in part, by different log-fold enrichment criteria used to identify cell-type-specific transcripts."

We are not sure whether any transcript is "exclusively" expressed in a tissue/cell type, but our work expands the list of "highly" enriched transcripts for the intestine. We note that it would theoretically be possible to identify medial and lateral cell subsets by subclustering, or by conducting scRNA-Seq on cells from these regions, as others have done for anterior and posterior tissues, etc. Nonetheless, our study demonstrates that spatial LCM is a straightforward way to achieve greater transcriptome depth when single-cell resolution is not required. We did not attempt to validate transcripts suggested to be intestine-enriched by scRNA-Seq studies, but not in our LCM data (>1000 transcripts in Figure 5C). However, it is now possible for others to use the revised Supplementary file 2 and/or the PlanGut website to identify these transcripts. As suggested by reviewers, some of these transcripts are in fact expressed in the intestine, but at lower fold-changes relative to non-intestinal tissue. We hope the new Results section and supplementary files more effectively facilitate comparison between our data and that of others.

Fourth, we also conducted Gene Ontology analysis on phagocyte, goblet cell, and basal cell expression data from Fincher et al., 2018 (Figure 6—figure supplement 2A) to develop a deeper understanding of the potential roles of each cell type. This analysis complements a new/additional GO analysis of the most medially (97) and laterally (56) enriched transcripts in our LCM data (Figure 6—figure supplement 2E), the majority of which are expressed in goblet cells.

Finally, we have shortened the text of the Gene Ontology and Human Protein Atlas analyses, and moved these sections to later in the manuscript, after all sections related to the number and location of cell types. These revisions have shortened the manuscript by ~900 words, and (in our opinion) improve the flow of the Results section.

We have also made a number of revisions to supplementary files to more clearly distinguish Supplementary files from Source data, and to facilitate easier interaction with expression data and comparison with scRNA-Seq data. A complete list of Supplementary files and Source data, with text legend descriptions, and detailed legends in each file, can be found at the end of the revised manuscript.

Individual reviews are show below for the authors to address minor points (editing for clarity).Reviewer #1:In the manuscript by Forsthoefel et al., the authors use laser capture microdissection with RNA sequencing to determine novel transcripts expressed in specific spatial locations in the planarian gut. The authors find an unappreciated spatial complexity of planarian gut gene expression, but also perform some comparative gene expression from the mammalian endodermal structures. The authors then dissect the function of a gli and RREB2 transcription factors in regulating goblet cell numbers. The manuscript is well-written and the experiments are of high quality. Below are a couple of concerns the authors might want to consider.1) On one hand, this manuscript is aiming to be a resource paper, and thus, it reads a bit like a laundry list of genes found. It is hard to put the current study in the context of previous ones, particularly the single-cell atlas-type papers. It would have been nice to contrast and compare what the authors found that the other studies missed, or why their method is better, etc.

We have revised supplementary files and developed a website, plangut.omrf.org, to enable more user-friendly interaction with our data. We believe this has made our in situ expression patterns much more accessible, as compared to displaying ~140 expression patterns in a figure supplement (which is still provided as Figure 2—figure supplement 1). Changes to supplementary files, along with the website, have also made it easier to compare and contrast our LCM results with previous scRNA-Seq studies. We have also prioritized putting our results in the context of previous studies by creating a stand-alone section in the Results ("Laser capture substantially increases resolution of the global intestinal transcriptome"), along with a main text figure (Figure 5). In addition, we used the Fincher scRNA-Seq data to add depth to our Gene Ontology (GO) analysis and discussion of possible functions for each cell type (Figure 6—figure supplement 2A). Lastly, we agree that the GO and Human Protein Atlas (HPA) analyses, on top of consideration of goblet-specific transcripts, made the text very "listy." We have trimmed the GO and HPA sections considerably, and placed most genes mentioned into Supplementary files for readers who are interested in these examples. The Results section is now ~900 words shorter; we hope the reviewers will find it more readable.

2) While the experiments and analyses are well done, it is hard to imagine the current study being widely used by planarian researchers or in the gut community as a whole. Part of the problem is that the data are not presented in a way that is easy for others to interact with them as a resource to come back to.

We thank the reviewer for this helpful advice. We hope the revisions to Supplementary files 1 and 2, and the PlanGut website make our data much more useful as a resource.

Reviewer #2:[…]1) I would suggest some clarification in the text about the description of the gli-1 and RREB2 RNAi phenotypes. In analysis of gli1(RNAi), the text in paragraph two of subsection “Intestine-enriched transcription factors regulate goblet cell differentiation and maintenance” highlights the observation that this treatment causes depletion of goblet cells within blastemas. However, it is also noteworthy that these cells seem reduced within the pre-existing regions as well and are also reduced in uninjured animals within the head and tail. This reading then sets up a comparison with the RREB2(RNAi) phenotype in which goblet cells from pre-existing regions were strongly reduced. However, RREB2(RNAi) goblet cells do not seem to have a normal abundance in the head blastema (Figure 7A/B), despite the description in the text ("arose normally"). I agree with the suggestion that the two genes could be subfunctionalized for generating versus maintaining goblet cells, but to me the evidence at hand does not present as much of an "all or nothing" case and so I find the term "mirror image" more a description of a model rather than definitively supported by the analysis. Though the acknowledgement of "complex observations" is appreciated here, some additional rewording could better align the text with the findings in the images or bring out these uncertainties. Alternatively, perhaps quantification or presenting other images from the animals probed could more clearly show the extent of distinctions between the phenotypes described in the text. Certainly, more experimentation on these two phenotypes could further clarify the function of the RREB2 and gli1 genes (eg, specificity for the intestine, effects on progenitor cells, direct tests of function on newly-formed versus pre-existing goblet cells, etc). However, I think it is beyond the scope of this report to embark on an extensive analysis of these two phenotypes at this time, as the work already makes a major contribution through the dataset for intestine cell transcriptomes, the LCM method, and the proof-of-principle use of the dataset for finding factors involved in intestinal biology.

We thank the reviewer for pointing out subtleties in these phenotypes that we did not adequately describe. The description of the *gli-1* and *RREB2* phenotypes in regenerates as "mirror images" of each was indeed an oversimplification, and we have deleted this sentence. In addition, we have modified this section in several places to more completely explain these important similarities and differences, as follows.

We have added the following text in the paragraph describing the *gli-1* phenotype:

"In addition, goblet cells were less abundant in pre-existing regions of the intestine, particularly in lateral intestinal branches (Figure 8A-B, Figure 8—figure supplement 2A-C)."

We have edited the paragraph describing the *RREB2* phenotype more extensively, reproduced here with changes underlined:

"By contrast to the *gli-1* phenotype, goblet cells appeared to differentiate normally in regenerating intestine upon knockdown of a third TF, *ras-responsive element binding protein 2 (RREB2),* including at the midline of anterior intestinal branches in tail fragments (Figure 8A-B; Figure 8—figure supplement 2A), in posterior branches in head fragments (Figure 8—figure supplement 2B), and in both anterior and posterior branches in trunk fragments (Figure 8—figure supplement 2C). However, in tail and trunk regenerates, new goblet cells were largely restricted to the midline/primary branches (Figure 8A-B, Figure 8—figure supplement 2A and 2C), and were less abundant in posterior branches of head fragments (Figure 8—figure supplement 2B). Furthermore, in pre-existing intestinal regions (especially lateral branches), goblet cell numbers were dramatically reduced or even completely absent. These included the posterior of tail fragments (Figure 8A-B; Figure 8—figure supplement 2A), the anterior of head fragments (Figure 8—figure supplement 2B), and central regions of trunk fragments (Figure 8—figure supplement 2C). As with *gli-1,* phagocytes and basal cells were unaffected (Figure 8A-B; Figure 8—figure supplement 2A-C). Together, these results suggest that *gli-1* regulates neoblast fate specification and/or differentiation of neoblast progeny into goblet cells in new intestinal branches, while *RREB2* maycontrol maintenance or survival of goblet cells after they initially differentiate. For both knockdowns, the reduction of goblet cells in lateral and pre-existing primary branches could be a consequence of reduced differentiation (*gli-1*) or maintenance/survival (*RREB2*) in these regions prior to amputation, during regeneration, or both."

We have added the following text to the Figure 8A Legend: "Both *gli-1(RNAi)* and *RREB2(RNAi)* also reduce goblet cells in lateral branches."

At the end of this section, we do indeed suggest potential avenues for future work to clarify the roles of these genes: "Thus, although our data support a role for *gli-1* and *RREB2* in goblet cells and/or their precursors, they also raise the possibility that basal cells (and possibly phagocytes or muscle cells) may non-autonomously influence goblet cell differentiation and/or survival."

2) The finding of shared intestinal expression between planarians and humans is very interesting. An aspect of this experiment that is less completely supported from the presentation was the assignment of gene orthology across these diverse species. For example, how were gene families handled in which there may not be a 1-to-1 orthology? (for example, Smed-piwi-1 top blastx hit in human genome is to HsHIWI, which itself reciprocally blasts to Smed-piwi-3 in the planarian genome, so presumably piwi-1 would not be present in the reciprocal best hits yet likely have some conserved molecular function in the piRNA pathway). Given the uncertainties in this process, including the orthology dataset would probably be the best way to allow readers to evaluate the strength of the orthology argument put forward here.

The Human Protein Atlas (Uhlén et al., 2015, https://doi.org/10.1126/science.1260419, assigned tissue enrichment scores based on individual transcripts/proteins, not on membership in a gene or protein domain family, which may have multiple members expressed in numerous tissues. Thus, determining whether the planarian intestine is similar to human digestive tissues at the transcriptome/proteome level requires some attempt to identify direct orthologs. Nonetheless, the reviewer rightly raises the central concern of sequence-based orthology inferences, including Gene Ontology: namely, that gene duplication, loss, and rearrangement necessitate approaches to ortholog/paralog identification that have inherent trade-offs ("precision-recall trade-offs," see Altenhoff et al., Nature Methods, 2016, https://doi.org/10.1038/nmeth.3830, and Altenhoff et al., Evolutionary Genomics, 2019, https://doi.org/10.1007/978-1-4939-9074-0_5). At risk of oversimplifying, stringent methods with high precision/few false positives (like the RBH/reciprocal best hit approach we used) potentially exclude orthologs/paralogs (false negatives), while other high recall methods with fewer false negatives result in more frequent incorrect ortholog/paralog identification (false positives). We felt that, for this initial analysis, a conservative approach (RBH) would be more appropriate. A comprehensive comparison of other tree- and graph-based methods for determining orthology and paralogy, while interesting, would be a considerable project of its own, since single planarian genes may have two or more human paralogs expressed in multiple tissues, and so on. Our RBH-based comparison, although not the only possible method, supports the more traditional gene ontology analysis in the previous figure, and offers an additional genome-wide perspective on homology between human and planarian digestive tissues, reinforcing the relevance of this emerging animal model.

Our orthology mappings between planarian and human were originally provided in Figure 3—source data 1, but we apologize that this was not clear from our minimal legend. We have now made these mappings available as Figure 7—source data 1 (E-H contain BLAST-based mapping and RBH identification data), separate from the list of planarian transcripts with orthologs enriched in human tissues (Supplementary file 4). We have also updated both legends to clearly identify what is available in each worksheet.

Reviewer #3:[…] The only concern I have in terms of impact is that whereas the optimised laser-capture method is widely applicable, the use for the intestinal gene expression dataset is largely limited to the planarian community.

At the moment, most of our knowledge of regeneration in the digestive tract is derived from mouse, *Drosophila*, and human organoids/cell lines. Over the long term, comparative analyses between these organisms and animals with more extensive regenerative capacity (e.g. planarians, sea cucumbers, salamanders, etc.) will help to develop a more comprehensive (and biomedically relevant) understanding of the similarities and differences in mechanisms animals use to promote regrowth of damaged digestive tissue. A first step is defining the cell types and states within digestive organs, which we conduct here. The revisions you suggest (making our data and comparisons with other studies more accessible in spreadsheets and a website, including sequences for all transcripts) will help to increase the broad applicability and utility of our study as a resource for others, and we are grateful for the suggestions.

The organization of the manuscript however should be improved. The text is long and reads as a long list of gene names grouped by expected function. Some of the functional processes even come by multiple times over the course of the descriptions of the various stainings. As there is no data verifying the actual involvement of the intestine in these processes or further analysis of mechanistic aspects, this information could be equally well presented in a table. The primary information seems to be in Figure 1 which shows the method and the analysis, and Figure 7 which contains some very interesting new biology. The other figures and the text describing them could and should be condensed.When turned into a short format, I think this is an interesting manuscript that makes a good case for laser-capture as a highly effective method to identify tissue-specific gene expression, and demonstrates that new biology can be uncovered by using it thoughtfully – as the authors clearly did.

We thank the reviewer for these useful suggestions. We agree that sections of the text, particularly those discussing Gene Ontology (GO) and Human Protein Atlas (HPA) comparisons, discussed too many genes. We have condensed and combined some of these sections, and moved most gene names to Supplementary files, as suggested. We also trimmed the number of genes discussed in the Results related to the FISH figures (now Figure 3, Figure 4, and Figure 3—figure supplement 1). These edits have reduced the length of the Results by ~900 words, and have also (we believe) improved the flow of the text.

However, we have left the GO and HPA figures in the main text. Even though function is inferred by gene homology, GO is still a widely used method for generating hypotheses about cell type function, and has also been used effectively to summarize region-specific functions in the digestive tract based on genome-wide expression data (for examples, see Marianes and Spradling, *eLife,* 2013, https://doi.org/10.7554/*eLife*.00886 and Buchon et al., Cell Reports, 2013, https://doi.org/10.1016/j.celrep.2013.04.001). In addition, our HPA comparison uses a conservative approach to identify homologs, is a unique demonstration of the conservation of digestive physiology in the planarian, and supports the relevance of this organism for studies of digestive system biology and regeneration. Similarly, we feel that our study uses WISH and FISH to more comprehensively characterize the number and locations of intestinal cell types and subtypes than previous efforts, and, in conjunction with the comparisons to scRNA-Seq data, is a useful advance. Accordingly, we have left these figures in the main body of the paper.